# Towards True Speech-to-Speech Models Without Text Guidance

**Xingjian Zhao**[1,3*] **Zhe Xu**[1,2,3*] **Luozhijie Jin**[1,2,3] **Yang Wang**[1,3]
**Hanfu Chen**[1,3] **Yaozhou Jiang**[1,3] **Ke Chen**[1,2,3] **Ruixiao Li**[1,2,3]
**Mingshu Chen**[1,3] **Ruiming Wang**[1,3] **Wenbo Zhang**[1,2,3] **Qinyuan Cheng**[1,3]
**Zhaoye Fei**[1,3] **Shimin Li**[3] **Xipeng Qiu**[1,2,3†]
[1]Fudan University
[2]Shanghai Innovation Institute
[3]MOSI.AI

## Abstract

Spoken dialogue systems often rely on cascaded pipelines that transcribe, process, and resynthesize speech. While effective, this design discards paralinguistic cues and limits expressivity. Recent end-to-end methods reduce latency and better preserve these cues, yet still rely on text intermediates, creating a fundamental bottleneck. We present a true speech-to-speech large language model that directly understands and generates speech without relying on text guidance. Our approach combines a modality-based layer-splitting architecture with a frozen pre-training strategy, preserving the reasoning and knowledge of pretrained text LLMs while adding native speech capabilities. Experiments show that our model achieves state-of-the-art results in spoken question answering and delivers comparable speech-to-speech performance relative to existing text-guided systems, while still maintaining competitive text performance. By narrowing the gap between text-guided and direct speech generation, our work establishes a new paradigm for expressive and efficient end-to-end speech interaction. We will release our code and models to support further research in true speech-to-speech foundation models.

## 1 Introduction

Speech is one of the most natural and intuitive modalities for human–computer interaction, making spoken dialogue systems a central focus of contemporary AI research. Traditional systems for spoken interaction are typically implemented using a cascaded pipeline: speech input is first transcribed into text, a text-based large language model (LLM) generates a response, and the output is subsequently converted into audio through a text-to-speech (TTS) module (Figure 1a). While this architecture leverages the full reasoning capacity of text-based LLMs, it inevitably discards information encoded in the original speech signal and constrains the system to produce only responses that can be faithfully represented in text.

Early end-to-end attempts such as GSLM (Lakhotia et al., 2021) and AudioLM (Borsos et al., 2023) demonstrated that speech could be modeled directly, but these works remained largely confined to experimental dialogue continuation tasks and faced challenges in scaling into full-featured assistants. Later work shifted toward text-guided generation as a compromise: SpeechGPT (Zhang et al., 2023a) used a chain-of-modality design, while Moshi (Défossez et al., 2024) and PSLM (Mitsui et al., 2024) achieved low-latency streaming through parallel speech–text generation. GLM-4-Voice (Zeng et al., 2024b) advanced this further by interleaving text and speech in chunk-based generation (Figure 1b), reaching near-text-level performance in streaming dialogue. Importantly, while GLM-4-Voice primarily relies on text-guided responses, it also supports direct speech generation—but its direct mode remains noticeably weaker than its text-guided counterpart.

---

*Equal contribution.
†Corresponding author. Email: zhaoxj24@m.fudan.edu.cn, chengqy21@m.fudan.edu.cn, xpqiu@fudan.edu.cn

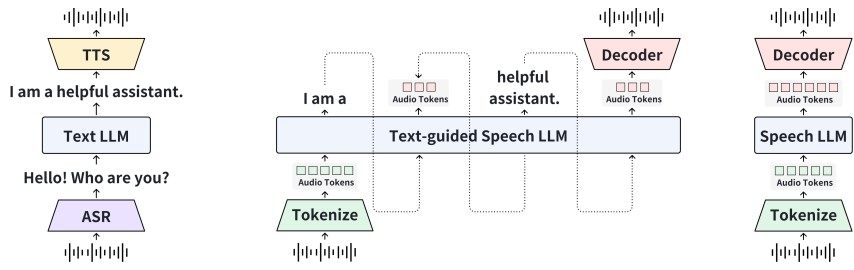

(a) Cascaded pipeline  (b) Text-guided speech models  (c) True speech-to-speech models

Figure 1: **Paradigms for spoken dialogue modeling.** (a) Cascaded pipelines rely on ASR → LLM → TTS, discarding paralinguistic cues. (b) Text-guided speech models incorporate speech input but still depend on text as an intermediate during generation. (c) True speech-to-speech language models directly comprehend and produce speech, avoiding the text bottleneck.

By accepting speech directly as input, these approaches preserve paralinguistic cues such as prosody, emphasis, and emotion. Yet their reliance on intermediate text during generation creates a fundamental bottleneck: it introduces latency, reduces efficiency, and restricts expressivity, since non-verbal vocalizations (e.g., laughter, hesitation) lack natural text equivalents. In addition, because of the inherent gap between speech and text modalities, current methods often introduce speech capability at the expense of text ability, leading to a measurable degradation in the backbone's text performance. For instance, SpiritLM(Nguyen et al., 2024) shows a notable drop in MMLU accuracy from 45.3 to 36.9 after incorporating speech modeling. Closing the gap between text-guided and direct speech generation is therefore critical for realizing true speech-to-speech interaction.

In this work, we introduce a novel approach that enables large language models to natively model speech while largely retaining their text-based capabilities. Our method builds on a pretrained text LLM backbone but diverges from prior approaches through a modality-specific layer-splitting scheme and a frozen pretraining strategy. This design preserves the backbone's linguistic knowledge while equipping the model with native speech understanding and generation abilities comparable to existing text-guided systems. As a result, our model can directly produce high-quality speech without relying on intermediate text representations, establishing a new paradigm for end-to-end speech-to-speech generation. Importantly, because the majority of knowledge remains in the pretrained text model, our approach avoids dependence on large-scale, knowledge-intensive speech datasets. Instead, alignment transfers reasoning, world knowledge, and generalization abilities from the text backbone to the speech modality.

The main contributions of this paper are as follows:

- We present a **true speech-to-speech large language model** that achieves **state-of-the-art** performance on speech-to-speech benchmarks without relying on any intermediate text guidance. At the same time, the model natively supports both text and speech as input and output modalities, thereby narrowing the gap between spoken and written interaction.

- We introduce **modality-based layer-splitting** and **frozen pre-training** that improves alignment between speech and text while mitigating the degradation of reasoning ability and world knowledge typically observed when extending LLMs to new modalities.

- We conduct extensive experiments and ablation studies to validate the effectiveness of our approach, demonstrating advanced speech–text cross-modal alignment and textual ability preservation.

## 2 MODEL ARCHITECTURE

For advancing toward a true speech-to-speech large language model, we add a modality-based layer-splitting to an autoregressive Transformer, enabling deep fusion of heterogeneous modalities and modality-specific generation.

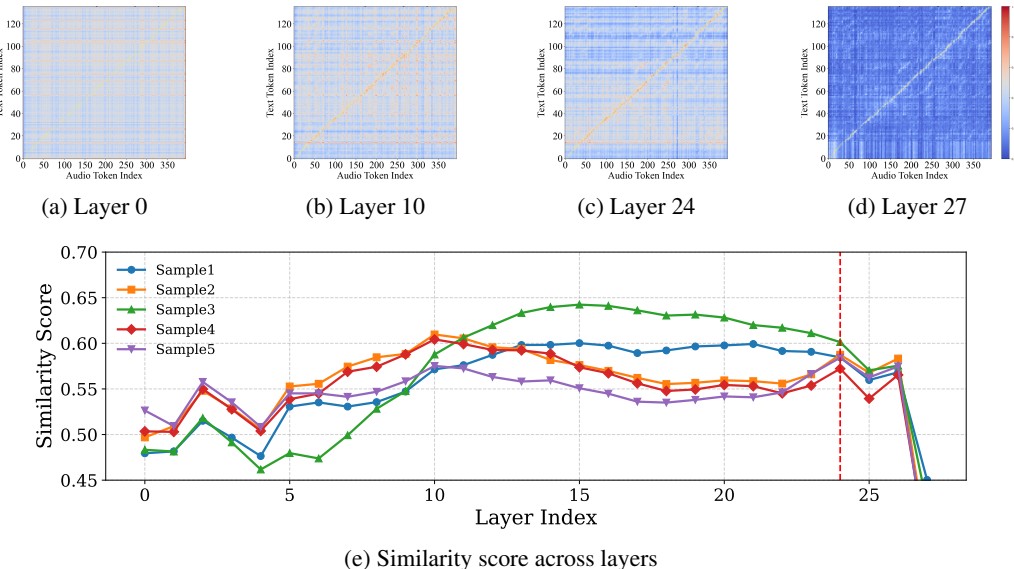

(a) Layer 0      (b) Layer 10      (c) Layer 24      (d) Layer 27

(e) Similarity score across layers

Figure 2: **Visualization of the layer-wise similarity between speech and text representations.** (a)–(d) Cosine similarity heatmaps at representative layers (0, 10, 24 and 27) reveal how cross-modal alignment evolves across the model depth. The yellow dots are the points selected by DTW sampling based on similarity. It can be seen that the points selected by our evaluation method largely coincide with the points of high similarity. The whole cosine similarity figure of 28 layers will be posted in Appendix D.1. (e) Similarity score across all layers on five samples shows a progressive increase up to around layer 10, then there are slight fluctuations in the subsequent 14 layers, followed by a noticeable decline in the final layers. This trend indicates that speech and text representations become gradually fused in the lower-to-middle layers but diverge again at the top layers. Content of samples are provided in Appendix D.2. More analyses across models are provided in the Appendix D.3.

.

## 2.1 MODALITY-BASED LAYER SPLIT

For the Transformer backbone, our design goal is to preserve the original text capabilities of LLMs while augmenting LLMs with speech understanding and generation. Existing approaches typically rely on the Depth Transformer (Défossez et al., 2024) that generates multiple VQ tokens as a single input, or alternatively, expand the vocabulary to directly encode speech tokens into the input sequence (Zeng et al., 2024b). However, our preliminary study on speechgpt2-preview(Open-Moss, 2025) revealed that the hidden-state alignment between a sentence and its corresponding speech sequence gradually deteriorates in deeper layers: while strong diagonal similarity emerges in lower layers, it vanishes in later layers.

As shown in the Figure 2, by examining the hidden-state similarity between the same spoken utterance and its corresponding text across different layers, we observe that in a 28-layer model, the similarity steadily increases in the first 11 layers, fluctuates and gradually stabilizes in the following 14 layers, and then decreases in the final 3 layers. This finding suggests that as the model is trained, the representations of text and speech become increasingly fused within the first 25 Transformer blocks, but gradually diverge in the last four layers.

Motivated by this, we introduce a modality-based layer split at the 32nd block of our 36-layer Transformer. At this point, the shared hidden state is routed into modality-specific branches: one branch continues through the final four layers to predict text tokens, while the other routes into a parallel four-layer stack to predict speech tokens.

This split-then-specialize design allows the model to leverage the first $N$ layers for joint multimodality fusion, while reserving the final layers for modality-specific generation. As a result, the archi-

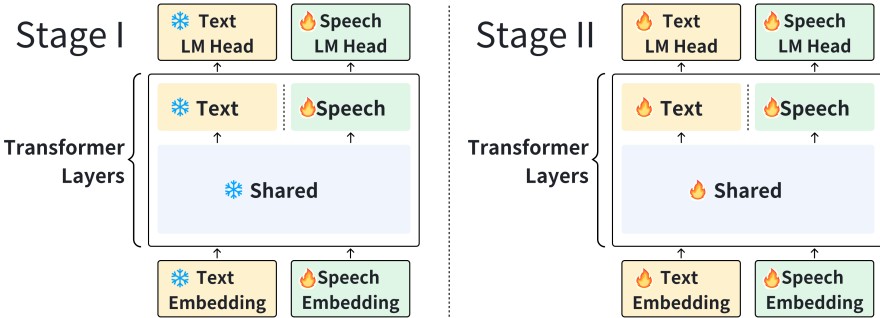

Figure 3: **Model architecture and training strategy.** We split the trailing Transformer layers based on modality, and freeze the text backbone during Stage I pre-training. Both branches are initialized from the same pretrained text model backbone.

tecture enhances cross-modality transfer, enabling the system to inherit the capabilities of textually-pretrained LLMs and express it natively in the speech modality.

## 2.2 SPEECH TOKENIZATION

Our speech tokenizer is designed with four key objectives: (1) to achieve a single-codebook, low-bitrate representation for efficient autoregressive generation and simplified context management; (2) to maximize semantic content in order to facilitate knowledge transfer from text to speech; (3) to preserve sufficient paralinguistic detail to enable faithful reconstruction of human speech; and (4) to support full streaming operation for low-latency processing.

**Encoder** Discrete speech tokenizers are commonly trained with reconstruction objectives (Gong et al., 2025; Zhang et al., 2023c) or self-supervised discovery methods (Shon et al., 2024; Liu et al., 2024). However, prior work has observed that tokens optimized primarily for reconstruction are often suboptimal for LLM learning (Défossez et al., 2024). To address this, and following the design of CosyVoice 2 (Du et al., 2024), we adopt automatic speech recognition (ASR) as the sole training objective for our tokenizer encoder. Our encoder is further trained based on the GLM-4-Voice Tokenizer (Zeng et al., 2024b), but we modify it to be fully causal rather than block-causal, thereby ensuring true streaming support.

**Decoder** For decoding, we adopt the flow-matching architecture introduced in CosyVoice 2 (Lipman et al., 2022; Du et al., 2024). While CosyVoice 2 employs chunk-attention to improve efficiency, this mechanism introduces undesirable time delays. To mitigate this issue, we compress the chunk size, which significantly reduces latency while maintaining reconstruction quality. This modification makes our tokenizer particularly well-suited for streaming dialogue systems that demand both high fidelity and low response delay.

## 3 TRAINING STRATEGY

### 3.1 PRE-TRAINING

The objective of pre-training is to introduce a speech modality into a pretrained text-based LLM while preserving its original text capabilities. To this end, we initialize our model from Qwen-3-8B(Yang et al., 2025) and adopt a two-stage pre-training strategy using a large-scale, high-quality speech corpus. The procedure is outlined below.

### 3.1.1 DATA COLLECTION AND PROCESSING

We begin with approximately 9 million hours of real-world audio data collected from the internet. To remove non-speech content, we apply a custom voice activity detection (VAD) pipeline based on `pyannote` (Plaquet & Bredin, 2023; Bredin, 2023), resulting in roughly 4 million hours of speech.

Table 1: Statistics of pre-training data. Hours are shown in thousands (k).

| Dataset | Total (h) | Real (h) | Synthetic (h) |
|---|---|---|---|
| English Interleaved | 690k | 624k | 66k |
| Chinese Interleaved | 952k | 876k | 76k |
| Unsupervised | 2,303k | 2,303k | 0 |

These data are organized into two categories according to source type: (1) *interleaved speech–text pre-training*, drawn primarily from podcasts, and (2) *unsupervised speech pre-training*, drawn primarily from video content. Podcasts are chosen for interleaved pre-training because they typically provide cleaner recordings and clearer speech, enabling automatic speech recognition (ASR) systems to generate more reliable transcripts. In contrast, video sources, while more diverse and noisier, are better suited for the unsupervised pre-training setting, where robustness to challenging acoustic conditions is essential.

For the interleaved task, we first apply automatic speech recognition (ASR) to obtain text transcripts. Connectionist Temporal Classification (CTC) word alignment is then used to segment the audio into random-length chunks of 3–6 seconds. Each chunk contains either the corresponding audio segment or its transcribed text, and sequences are constructed by interleaving the two modalities. For unsupervised speech pre-training, we simply use full-length audio segments without transcript.

To mitigate the low knowledge density inherent in natural speech corpora, we also synthesize additional interleaved data from high-quality text corpora. Following the approach of Zeng et al. (2024c), we use FineWeb-Edu (Lozhkov et al., 2024) for English and Chinese FineWeb-Edu V2.1(Yu et al., 2025) for Chinese. These texts are converted into audio using the CosyVoice 2 TTS system (Du et al., 2024), producing large-scale synthetic speech–text pairs that enrich the training corpus.

A summary of dataset statistics is provided in Table 1.

### 3.1.2 TWO-STAGE PRE-TRAINING

We initialize our model from the Qwen3-8B backbone and employ a two-stage pre-training pipeline designed to introduce the speech modality while preserving the model's text capabilities.

**Stage 1: Speech Alignment with Frozen Text Backbone**   In the first stage, we freeze all parameters of the Qwen-3-8B backbone and train only the newly introduced speech-related components, including the speech token embeddings, speech-specific transformer layers, and the speech language modeling (LM) head. This stage serves to initialize speech parameters and establish stable alignment with the pretrained text representations. Training is conducted for approximately one epoch using the AdamW optimizer with cosine learning rate scheduling. The initial learning rate is set to $4 \times 10^{-4}$, with a batch size of 2.2M tokens, weight decay of 0.1, context length of 14,336 tokens.

**Stage 2: Joint Training with Text Knowledge Preservation**   In the second stage, we unfreeze a larger portion of the model to allow cross-modal adaptation. We experiment with three configurations: (1) unfreezing the entire model and training all parameters jointly, (2) unfreezing only the shared transformer layers while keeping the text embeddings, text-specific layers, and text LM head frozen, and (3) gradually unfreezing the shared layers in reverse order (from last to first). Since unfreezing text parameters risks degradation of textual abilities, we incorporate additional text-only pre-training data to preserve the model's linguistic competence. Specifically, we include FineWeb-Edu (Lozhkov et al., 2024) for English and Chinese FineWeb-Edu V2.1 (Yu et al., 2025) for Chinese, filtering entries with quality scores $\geq 3$.

Stage 2 training is conducted for two epochs on the same speech dataset used in Stage 1, combined with 0.1 epoch of text-only pre-training data. Hyperparameters are largely consistent with Stage 1, except that the learning rate is reduced (decaying from $6 \times 10^{-5}$ to $6 \times 10^{-6}$) and the batch size is increased to 2.8M tokens to account for the additional text data. In practice, the three configurations achieve comparable results. For simplicity, we adopt configuration (1) as the default initialization for subsequent supervised fine-tuning. A detailed ablation study is provided in section 5.

## 3.2 SUPERVISED FINE-TUNING

### 3.2.1 DATA ADAPTATION AND CONSTRUCTION

Because high-quality supervised fine-tuning data for speech assistants are scarce in natural settings, we construct such data synthetically. Our process begins with existing open-source text-based supervised fine-tuning datasets listed in Appendix B.

**Text Adaptation**   We employ the GPT-5 API to transform question–answer pairs into formats suitable for speech representation. This process involves (i) converting non-vocal content such as mathematical expressions, tables, or Markdown into TTS-compatible forms, and (ii) filtering out instances that cannot be effectively rendered as speech (e.g., long code dumps or dense LaTeX passages). Adaptation also improves data quality by shortening excessively long responses to make them more appropriate for spoken delivery, correcting obvious factual errors, and suggesting suitable emotional tones for TTS synthesis. Prompt for the adaptation process is available in Appendix C.

**Speech Synthesis**   The adapted text is then synthesized into audio using multiple TTS systems. We primarily employ Seed-TTS (Anastassiou et al., 2024) from VolcEngine. For the *user role*, we generate speech with a diverse set of speaker voices to improve robustness. For the *assistant role*, we always use a single consistent speaker to establish a stable and recognizable system identity. To further enhance voice diversity, naturalness, and stylistic control, we additionally employ MOSS-TTSD (Team, 2025) to synthesize conversational datasets. By assigning different system prompts, we can vary the assistant's speaking style and role in a controllable manner.

**Quality Filtering**   Although LLM-based TTS systems produce highly natural speech, they are susceptible to synthesis errors. To mitigate this, we apply automatic quality filtering using SenseVoice-Small ASR (An et al., 2024). Specifically, we discard entries whose ASR transcripts exhibit a word error rate (WER) $\geq 0.2$ relative to the original text.

**Statistics**   In total, we end up with over 1500k question–answer pairs for supervised fine-tuning, comprising approximately 650k English pairs and 860k Chinese pairs.

### 3.2.2 TRAINING DETAILS

Building on the pretrained model, we conduct supervised fine-tuning on the constructed multimodal dataset for two epochs. Training is performed with the AdamW optimizer, using a cosine learning rate schedule that decays from $1 \times 10^{-5}$ to $1 \times 10^{-6}$. We use a batch size of 8, apply a weight decay of 0.1, and set the maximum context length to 10,240 tokens with sequence packing.

To further strengthen cross-modal alignment between speech and text, fine-tuning incorporates four input–output modality configurations: *speech question → speech answer*, *speech question → text answer*, *text question → speech answer*, and *text question → text answer*. The modality pairing is controlled by system prompts, while the underlying content remains identical across configurations. This design ensures that the model learns to handle both unimodal and cross-modal interactions, enabling it to accept text or speech as input and generate either text or speech as output within a unified framework.

## 4 EVALUATION

### 4.1 TOKENIZER

In this section, we present the experimental evaluation of our encoder and decoder components. For the encoder, a crucial aspect is the preservation of semantic information. To assess this, we fine-tuned a Qwen3-0.6B model (Yang et al., 2025) for ASR. Distinct from embedding-based approaches (Yang et al., 2021), our method directly leverages discrete codebook IDs generated by various encoders as input, better aligning with the Large Language Model (LLM) paradigm. This ASR model was trained on the 960-hour Librispeech training dataset (Panayotov et al., 2015). We then evaluated the corresponding Word Error Rate (WER) on the test sets (test-clean, test-other, and dev-clean). Each model was trained for 100k steps using a batch size of 128 and a learning rate of

1e-4, and we report the lowest WER achieved. Our baselines include codecs designed to capture semantic information, such as Mimi (Défossez et al., 2024) and XCodec 2.0 (Ye et al., 2025), as well as ASR-trained codecs like GLM-4-Voice (Zeng et al., 2024a), CosyVoice (Du et al., 2024) and CosyVoice 2 (Du et al., 2024). Ours represent our proposed streaming model, which is further fine-tuned from GLM-4-Voice.

Table 2: Evaluation results of our speech encoder

| Model | Frame Rate (Hz) | BPS | Streaming | WER (%) ↓ | | |
|---|---|---|---|---|---|---|
| | | | | test-clean | dev-clean | overall |
| Mimi-8 | 12.5 | 1100 | × | 9.65 | 9.67 | 14.45 |
| XCodec2.0 | 50 | 800 | × | 14.17 | 13.82 | 20.07 |
| Cosyvoice | 25 | 300 | × | 10.15 | 9.64 | 14.21 |
| Cosyvoice2 | 25 | 325 | × | 9.45 | 9.42 | 13.78 |
| GLM-4-Voice | 12.5 | 175 | Chunk(2s) | **6.59** | **6.07** | **9.17** |
| Ours | 12.5 | 175 | ✓ | 7.89 | 7.29 | 10.80 |

To evaluate our decoder, we utilize the Seed-TTS-Eval benchmark (Anastassiou et al., 2024), employing its standard English and Chinese test datasets. We assess intelligibility (WER), speaker similarity (SIM), and speech quality (DNSMOS). Speaker similarity (SIM) is computed as the cosine similarity between WavLM-TDNN embeddings (Chen et al., 2022) of the prompt and generated speech. WER is measured using whisper-large-v3 (Radford et al., 2023) for non-Chinese languages and paraformer-zh for Chinese (Gao et al., 2022). Additionally, we incorporate the DNSMOS (Reddy et al., 2022) metric to assess the perceived quality of the generated speech. Since our decoder is fine-tuned from CosyVoice 2, we compare it directly with the CosyVoice series.

Table 3: Evaluation results of our speech decoder

| Model | Frame rate | Seed-TTS-Eval-EN | | | Seed-TTS-Eval-ZH | | |
|---|---|---|---|---|---|---|---|
| | | WER ↓ | SIM ↑ | DNSMOS ↑ | WER ↓ | SIM ↑ | DNSMOS ↑ |
| Cosyvoice | 25hz | 10.53 | 0.66 | 3.07 | 11.29 | 0.74 | 3.21 |
| Cosyvoice2 | 25hz | 4.63 | **0.68** | 3.09 | 3.11 | **0.75** | 3.22 |
| Ours | 12.5hz | **4.14** | 0.67 | **3.10** | **2.86** | 0.73 | **3.24** |

Our experimental evaluation demonstrates the robust performance of Our codec across both encoder and decoder components. For the encoder, Ours achieve an overall Word Error Rate (WER) of 10.80%. While this is slightly higher than the 9.17% of GLM-4-Voice, it is important to note that GLM-4-Voice operates with 2-second processing blocks rather than pure streaming. While being a full streaming architecture, Our model achieves a competitive WER. Furthermore, Our encoder significantly surpass other non-streaming codecs like Mimi-8 (14.45%) and CosyVoice 2 (13.78%), despite its lower BPS and frame rate. Consequently, our decoder, fine-tuned from CosyVoice 2, benefits from this enhanced capture of semantic information. Even at a lower frame rate, Our decoder achieves better intelligibility (lower WER) and perceived speech quality on both English and Chinese benchmarks compared to CosyVoice 2, with only a marginal trade-off in speaker similarity.

## 4.2 PRE-TRAINING

To assess the effectiveness of our pre-training strategy, we evaluate the resulting speech-enabled model on both speech modeling and text understanding benchmarks.

For *speech modeling ability*, we use StoryCloze (Hassid et al., 2023) together with our in-house Chinese counterpart, zh-StoryCloze. These benchmarks test the model's capacity to reason over and generate coherent speech continuations.

For *textual capability preservation*, we evaluate on MMLU (Hendrycks et al., 2021b;a) and CMMLU (Li et al., 2024), which measure knowledge and reasoning across diverse subject domains.

Table 4: **Evaluation result of our pre-trained model.** In the table, "S.C." refers to "StoryCloze", "s" refers to "Spoken", "t" refers to "Topic". SpiritLM results are takens from Nguyen et al. (2024). The tS.C. and sS.C. results for GLM-4-Voice are taken from Zeng et al. (2024b), and the tS.C. and sS.C. results for Moshi are taken from Défossez et al. (2024). Chinese language evaluations are not performed on models trained only in English.

| Model | Speech | | | | Text | |
|---|---|---|---|---|---|---|
| | tS.C. | sS.C. | zh-tS.C. | zh-sS.C. | MMLU | CMMLU |
| Moshi | 83.60 | 62.70 | - | - | 49.8 | - |
| GLM-4-Voice | 82.90 | 62.40 | 83.27 | 69.10 | 57.49 | 54.39 |
| SpiritLM | 82.90 | 61.00 | - | - | 36.90 | - |
| Ours | **84.87** | **63.17** | **90.32** | **71.94** | **67.19** | **69.53** |

Table 5: **Spoken question answering evaluation results & speech quality.** L./T./W. QA refer to LlamaQA, TriviaQA, and WebQA, respectively. Except for our model and GLM-4-Voice*, results for other models in the table are taken from Zeng et al. (2024b) and Défossez et al. (2024). We follow KimiTeam et al. (2025) to normalize the answer before judging.

| Model | L. QA | | T. QA | | W. QA | | UTMOS |
|---|---|---|---|---|---|---|---|
| | $S \to T$ | $S \to S$ | $S \to T$ | $S \to S$ | $S \to T$ | $S \to S$ | |
| **Pre-trained Model** | | | | | | | |
| GLM-4-Voice | 64.70 | 50.70 | 39.10 | 26.50 | 32.20 | 15.90 | - |
| TWIST | - | 4.00 | - | - | - | 1.50 | - |
| **Supervised Fine-tuned Model** | | | | | | | |
| SpeechGPT* | - | 21.60 | - | 14.80 | - | 6.50 | 4.00 |
| Moshi | - | 21.00 | - | 7.30 | - | 9.20 | 2.81 |
| Moshi* | - | 62.30 | - | 22.80 | - | 26.60 | - |
| GLM-4-Voice* | 74.33 | 65.67 | 45.90 | 43.20 | 39.22 | 38.34 | 4.25 |
| Ours | **77.33** | **63.67** | **45.20** | **28.80** | **45.90** | **36.71** | **4.37** |

\* : $S \to S$ results obtained with text guide

This dual evaluation allows us to verify that (i) the model acquires robust speech modeling abilities, while (ii) maintaining the original linguistic competence of the pretrained text backbone.

### 4.3 SUPERVISED FINE-TUNING

To comprehensively evaluate the capabilities of our SFT model, we assess QA ability using **LLaMA-Question**, **Trivia QA**, and **Web Questions** (Nachmani et al., 2024; Joshi et al., 2017; Chang et al., 2022). And the quality of generated speech is evaluated with **UTMOS** (MOS style evaluation) (Saeki et al., 2022).And we additionally conducted a subjective evaluation experiment, relevant details are provided in the Appendix H.

## 5 ABLATION STUDY

We study the effect of two key components in our pre-training pipeline: *Modality-based Layer Split* and *Frozen Pre-training*.

We first compare a naive baseline without either strategy (NF–NoSplit) against a variant that introduces *Modality-based Layer Split* but trains all parameters directly (NF), isolating the benefit of modality separation. Next, we evaluate the effect of *Frozen Pre-training* by comparing NF with FP–Full, where text parameters are frozen during pre-training and then unfrozen.

Table 6: **Ablation study on pre-training strategy.** FP: Frozen Pretrain (text parameters frozen during pretrain). **FP–Full**: all parameters unfrozen after Frozen Pretrain. **FP–Layerwise**: shared layers gradually unfrozen from last to first. **FP–Shared**: only speech–text shared layers unfrozen, text-specific remain frozen. **NF**: No Frozen Pretrain (all parameters trained directly). **NF–NoSplit**: NF without *Modality-Based Layer Split*, i.e., speech tokens added directly into text vocab without modality-specific layers. All models are trained for around 2 epochs on the pre-training dataset.

| Model | Split Layers | Speech | | | | Text | |
|---|---|---|---|---|---|---|---|
| | | tS.C. | sS.C. | zh-tS.C. | zh-sS.C. | MMLU | CMMLU |
| FP–Full | 4 | **85.20** | 63.12 | **90.21** | 72.10 | 66.50 | 69.15 |
| FP–Layerwise | 4 | 84.77 | 62.64 | 90.11 | 71.51 | **68.82** | 69.26 |
| FP–Shared | 4 | 83.27 | **63.50** | 90.11 | **72.69** | 67.26 | **69.27** |
| NF | 8 | 78.73 | 56.33 | 88.88 | 69.16 | 62.92 | 63.84 |
| NF | 6 | 79.05 | 56.49 | 89.10 | 67.93 | 63.27 | 63.79 |
| NF | 4 | 77.66 | 56.60 | 88.51 | 67.56 | 62.11 | 64.11 |
| NF | 2 | 78.09 | 56.87 | 88.62 | 68.31 | 62.92 | 63.74 |
| NF–NoSplit | 0 | 77.12 | 55.80 | 88.72 | 67.02 | 60.97 | 63.73 |
| Qwen3-8B | - | - | - | - | - | 76.60 | 77.35 |

We further ablate different unfreezing strategies after Frozen Pre-training: (i) FP–Full, unfreezing all parameters at once; (ii) FP–Shared, unfreezing only speech–text shared layers while keeping text-specific parameters frozen; (iii) FP–Layerwise, gradually unfreezing shared layers from last to first. The learning rate schedule for FP–Layerwise is described in Appendix F.

Results in table 6 highlight three main findings: (1) *Modality-based Layer Split* improves both speech modeling and textual ability preservation; (2) *Frozen Pre-training* provides substantial additional gains; (3) unfreezing strategies yield relatively small differences.

Overall, the ablation confirms that modality separation and freezing text parameters during pre-training are both critical to balancing speech learning with text knowledge preservation. While different unfreezing schedules provide slight trade-offs, their impact is minor compared to the gains from *Modality-based Layer Split* and *Frozen Pre-training* themselves.

## 6 RELATED WORKS

A detailed discussion of related work is provided in Appendix G.

## 7 CONCLUSION

We introduced a large language model capable of true speech-to-speech interaction without intermediate text, advancing the state of spoken dialogue systems beyond cascaded and text-guided frameworks. Our modality-based layer-splitting and frozen pre-training strategies enable the transfer of linguistic and reasoning knowledge to speech modality from pretrained text LLMs while preserving text abilities, avoiding the degradation often observed in multimodal adaptation. Our model achieves state-of-the-art results in spoken question answering, while supporting both text and speech as native input and output modalities. This work demonstrates that end-to-end speech modeling can reach near parity with text-guided methods while overcoming their inherent limitations in latency and expressivity. Looking forward, we envision speech-native models as the foundation of future human–AI interaction, supporting seamless, multimodal dialogue across diverse languages and contexts.

## ETHICS STATEMENT

This work introduces a speech-enabled large language model. While the model has potential benefits for accessibility and natural human–computer interaction, it also carries risks. Despite incorporating

alignment datasets during fine-tuning, the model may still produce unsafe or biased content. Additionally, the speech decoder could be misused for voice cloning or impersonation. We do not release tools or data optimized for such misuse and recommend responsible deployment with safeguards to mitigate these risks.

## ACKNOWLEDGMENTS

This work was supported by the National Natural Science Foundation of China (No. U24B20181), Shanghai Pilot Program for Basic Research - Fudan University 21TQ1400100 (22TQ018), and Fudan Kunpeng & Ascend Center of Cultivation.

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

## A    LLM Usage Disclosure

ChatGPT 5 was used to refine the writing style of certain paragraphs and as a supplementary tool to suggest related works. It was not the sole or primary source for the related work section; all references were independently identified, reviewed, and selected by the authors. The LLM did not contribute to research design, experiments, analysis, or results. The authors assume full responsibility for the content of this paper.

## B    Supervised Fine-tuning Datasets

Table 7 lists the supervised fine-tuning datasets used in our work. We report only the number of examples actually used for training.

| Dataset | Language | Used Samples |
|---|---|---|
| OpenHermes-2.5 | EN | 200k |
| OpenHermes-2.5 (Chinese translated) | ZH | 200k |
| Magpie-Llama-3.1-Pro-MT-300K-Filtered | EN | 300k |
| Magpie-Qwen2-Pro-200K-Chinese | ZH | 200k |
| BAAI_OL-CC | ZH | 11.7k |
| RefGPT-Fact | ZH | 50k |
| COIG-CQIA | ZH | 45k |
| Ruozhiba | ZH | 1.4k |
| Huatuo26M-Lite | ZH | 30k |
| Align-Anything-Instruction-100K | EN&ZH | 100k |
| Chinese-DeepSeek-R1-Distill-SFT | ZH | 110k |
| Chain-of-Thought-ShareGPT | EN | 7.14k |

Table 7: Supervised fine-tuning datasets used in our experiments.

## C    Prompt for Supervised Fine-tuning Text Adaptation

```
You are converting a supervised fine-tuning dataset (Q&A) into a format
    that can be read naturally by a TTS system for a speech language
    model. For each question and answer pair, you must determine whether
    the text is suitable for TTS according to the following rules.
    Possible states are: require no changes (PASSTHROUGH), requires
    adaptation (ADAPT), or unsuitable (REJECT).

CORE RULES
1) Preserve Meaning, Adapt for Speech
   - Preserve content verbatim where possible, but always rephrase rigid
   or written phrasing into natural, flowing speech transcript.
   - Adapted transcript should not include parentheses, brackets,
   symbols, Markdown, or formatting that only works on paper, or
   structured written formatting like "Method:... Result:...".
   - Do NOT add meta-statements such as "the spoken version is".

2) Convert Non-Vocal Elements
   - Math & Formulas: simple inline LaTeX is supported by TTS. Convert
   all math to inline LaTeX, e.g. $\cos 30^\circ =\frac {\sqrt
   {2}}{2}$. Complex or block LaTeX (derivatives, integrals, etc.) is
   not supported, read out if feasible, otherwise REJECT. Note that
   inline LaTeX command involving summations/limits,
   integrals/derivatives, sets/intervals, vectors/matrices, Greek
   letters (except \pi), \approx, \text, etc. are not supported. Use
   English letters instead of Greek letters for variables. [IMPORTANT]
   Single variables (e.g., $x$) MUST be wrapped in LaTeX.
   - Tables: content summary, e.g., "First, two CPUs at two hundred
   dollars each, then one GPU at one thousand dollars".
```

```
     - URLs: rephrase or remove. Only simple URLs may be read ("google dot
     com").
     - Short Code: narrate, e.g., "this.parseOptions": "this dot parse
     options".
     - Other non-vocal elements: narrate or rephrase.
     - Do NOT explicitly read out punctuation.

3) Length Constraint
        - If the content exceeds 200 words, shorten it to ~200 words
     while preserving factual correctness and logical flow.
        - If the question explicitly requests detail, allow up to 400
     words.
        - Prioritize clarity over brevity – it is acceptable to keep
     light redundancy or slightly exceed the word limit if it improves
     understanding.
     - If simplification cannot be done without major distortion, REJECT.

4) Language Policy
     - Supported languages: English and Simplified Chinese.
     - Mixed English-Chinese allowed. Do not translate.
     - Unless specified, avoid Classical Chinese or Old English.
     - If other languages appear: REJECT.

5) Formatting-Specific Prompts
     - If the question requires a format impossible for speech (tables,
     LaTeX \boxed{}), REJECT if inseparable from meaning.
     - Otherwise, keep the question unchanged and begin the answer with a
     polite clarification: "Sorry, I cannot provide the answer in a
     table. However..."
     - Do NOT add meta-statements like "this is the adapted answer."

6) Pause & Rhythm Control
     - Adjust punctuation or sentence boundaries to guide natural pauses
     and rhythm.
     - Break long or complex sentences into shorter ones.
     - Insert natural discourse markers when helpful ("so," "in other
     words," "for example").

7) Oral Smoothness
     - Always ensure the adapted text sounds like something a human would
     naturally say aloud.
     - Favor conversational flow, short clauses, and rhythm over rigid
     literalism.
     - Add light redundancy for clarity if needed ("That means...," "In
     short...").
     - Prioritize spoken fluency over textual fidelity.

8) Correction
     - If the answer contains incorrect or irrelevant content, correct it.
     - Do not reject unsafe or inappropriate content. Instead, rewrite the
     answer to make it appropriate.
     - Unless specifically instructed, the answer should be neutral and
     objective. Rewrite answers that are not.
     - Keep corrections minimal and faithful to the question. Do NOT add
     meta statements like "the corrected answer is".

9) Chain of Thought
        - When answering a complex question from the user: if the
     assistant gives the answer first, then reasons, reorder to reasoning
     first, then answer. (Unless explicitly instructed otherwise.)

10) Style
     - Recommend a TTS style for each content.
     - Available options: neutral, happy, sad, angry, surprised, fear,
     hate, excited, coldness.
```

```
11) Rejection Policy
    - REJECT if content contains long code dumps, complex math, dense
     LaTeX, giant tables, etc.
    - REJECT if any of the rules above are violated.

OUTPUT RULES
Return a JSON that follows this schema:
{
  "question": {
    "state": "PASSTHROUGH | ADAPT | REJECT",
    "text": "",        // adapted text if ADAPT, else empty
    "style": "",       // recommended TTS style
    "simplified": false
  },
  "answer": {
    "state": "PASSTHROUGH | ADAPT | REJECT",
    "text": "",
    "style": "",
    "corrected": false,
    "simplified": false
  },
  "quality": 1-5     // 5 = excellent, 1 = unusable
}
```

# D    SIMILARITY DETAILS

## D.1    HEATMAPS

Figure 4 shows the similarity maps of all layers for sample 0. It can be observed that the similarity diagonal begins to appear at layer 7, becomes clearly noticeable by layer 11, and after layer 24, the similarity of other tokens gradually increases at layers 25 and 26, while at layer 27, all similarities drop significantly. This pattern is consistent with the observations in our Similarity Score figure.

## D.2    SAMPLES

Below are the five random samples we used to calculate the similarity score, with all text content aligned and presented in English.

```
Sample0:
    Speech-to-text alignment is a core problem in the field of speech
    processing, and it becomes especially critical during the training
    of large-scale speech models. "Alignment" refers to accurately
    matching acoustic segments of the speech signal with characters,
    words, or subword units in the text sequence along the temporal
    dimension, enabling the model to learn the mapping between speech
    and language. If the alignment is inaccurate, the model struggles to
    capture the correspondence between speech and text effectively,
    which in turn affects the performance of speech recognition, speech
    synthesis, and multimodal tasks. Therefore, building a high-quality
    speech-text alignment mechanism is not only a fundamental step in
    training large speech models but also a prerequisite for enhancing
    model generalization and practical performance.

Sample1:
    In the previous lesson, we learned the concepts of the greatest
    common divisor (GCD) and the least common multiple (LCM), as well as
    how to calculate them. Today, we will continue to learn about the
    concept of prime numbers and use prime factorization to find the
    common divisors of two numbers.
    First, let us review what we learned in the previous lesson. We
    mainly studied several methods to find common divisors. The first
```

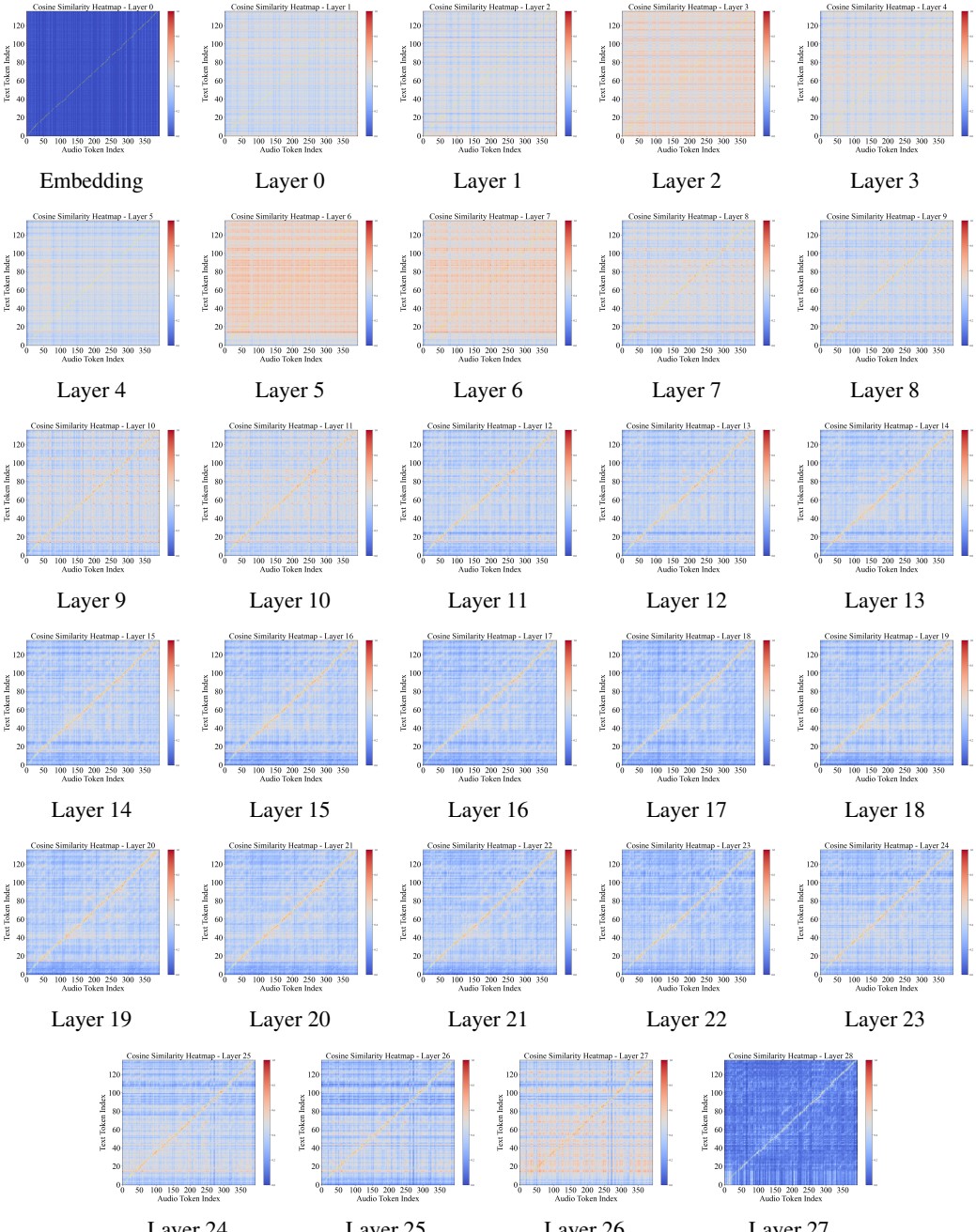

Figure 4: Heatmaps for embedding and layers 0–27.

```
    method is the listing method. The listing method is a general
    approach that can be applied to any two numbers

Sample2:
    A team of emerging scientists has developed a novel artificial
    intelligence algorithm that, by observing and analyzing human brain
    activity, successfully replicates the thinking processes of the
    human brain. This enables AI to simulate human thought while
    performing computations and processing information more efficiently,
    generating a significant response in the global scientific
    community. According to reliable sources, the research findings have
    already passed review by international academic institutions and
    will be presented and announced this month at the world's most
    prestigious scientific conference. It is expected to have a profound
    impact on the development of artificial intelligence.

Sample3:
    What about large language models? How do they work? They learn from
    an enormous corpus of text using natural language processing to
    understand the relationships between all the sentences in that
    corpus. For example, if one person says a sentence and another
    person responds, there is often a certain relationship between the
    two sentences. For instance, if the first person says, "I'm hungry,"
    the second person might respond, "I can make something for you."
    Once the model learns these relationships, it can use them to
    perform tasks such as translation or generating appropriate
    responses in other contexts.

Sample4:
    The harsh way of survival. Some species rely on complex social
    structures or specialized physical traits to ensure their survival
    and reproduction. Some species can even assist humans in locating
    resources. In the tropical regions of the Pacific, certain species
    live on the water surface, while others inhabit the seafloor. There
    are also species whose range spans both the surface and the
    seafloor, forming what is known as a three-dimensional ecosystem.
    Due to the vast expanse of the Pacific, these species must adapt to
    survive.
```

### D.3 Layer-wise Similarity Score of Different Models.

We conducted layer-wise similarity analyses on more models and a larger set of samples. For the two models shown in the Figure 5, due to implementation constraints that prevent us from mapping text tokens back to the original text, we used a global DTW–based metric to measure similarity score. In contrast, for the Figure 6, we adopted the fine-grained similarity computation method described in Appendix E. In both cases, we used the same set of 1K long-form speech samples and averaged the scores to obtain each model's layer-wise similarity profile.

From the results, we observe a consistent trend across all speech models: similarity increases from shallow to middle layers, and then decreases toward the deeper layers. In the global-DTW setting, the last few layers show a slight increase because stopwords and punctuation often lead to artificially higher similarity. However, this does not alter the overall trend of "rising then falling."

Across all models in the figure, a unified pattern can be observed: similarity begins to decline at approximately one-third of the total depth. The above analysis result guided our model design choice.

## E Layer-wise Similarity Score.

To evaluate cross-modal alignment inside a Transformer backbone, we compute a similarity score at each layer $i \in \{1, \ldots, L\}$. Let the text tokens be $\{t_1, \ldots, t_n\}$ and the speech tokens $\{s_1, \ldots, s_m\}$.

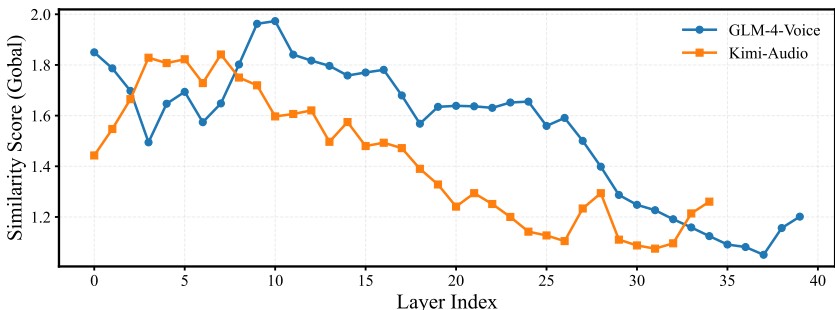

Figure 5: **Similarity Score of GLM-4-Voice and Kimi-Audio.** Due to implementation constraints that prevent us from mapping text tokens back to the original text, we used a global DTW–based metric to measure similarity score in this figure.

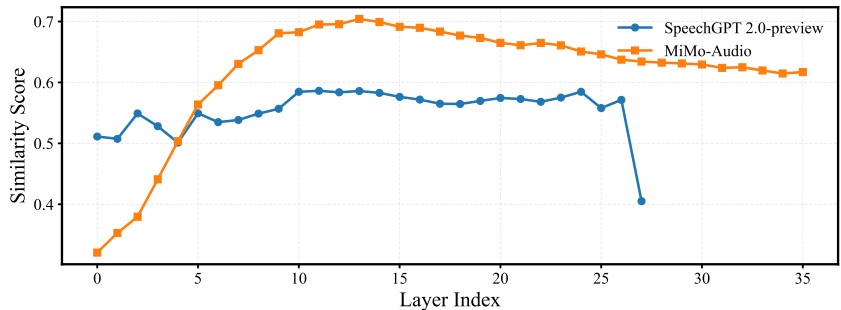

Figure 6: **Similarity Score of SpeechGPT 2.0-preview and MiMo-Audio.** We adopted the fine-grained similarity computation method described in Appendix E.

Using forced alignment(Pratap et al., 2023), we obtain $J_i$ alignment pairs

$$\{(T_{i,1}, S_{i,1}),\ (T_{i,2}, S_{i,2}),\ \ldots,\ (T_{i,J_i}, S_{i,J_i})\},$$

where $T_{i,j}$ is the set of aligned text tokens and $S_{i,j}$ is the corresponding set of speech tokens (the alignment is fixed, but hidden states depend on the Transformer layer $i$).

For each pair $(i, j)$ we construct a cosine similarity matrix

$$M_{i,j}[u, v] = \cos\left(h_{i,t_u},\ h_{i,s_v}\right),$$

where $h_{i,t}$ and $h_{i,s}$ denote hidden states of text token $t$ and speech token $s$ at layer $i$.

The DTW-based similarity is defined as

$$\mathrm{DTW}_{i,j} = \frac{1}{|P_{i,j}|} \sum_{(u,v)\in P_{i,j}} M_{i,j}[u, v],$$

where $P_{i,j}$ is the optimal Dynamic Time Warping path through $M_{i,j}$, i.e., the alignment trajectory maximizing similarity under temporal constraints.

As a background normalization, we compute

$$\mathrm{BG}_{i,j} = \frac{1}{|S_{i,j}|\,(n - |T_{i,j}|)} \sum_{s\in S_{i,j}} \sum_{t\in\{t_1,\ldots,t_n\}\setminus T_{i,j}} \cos\left(h_{i,t},\ h_{i,s}\right).$$

Finally, the **layer-wise similarity score** at layer $i$ is

$$\mathrm{SS}_i = \sum_{j=1}^{J_i} \left(\frac{\mathrm{DTW}_{i,j}}{\mathrm{BG}_{i,j} + \lambda}\right), \qquad \lambda = \frac{1}{\sum_{k=1}^{L} J_k} \sum_{k=1}^{L} \sum_{j=1}^{J_k} \mathrm{DTW}_{k,j}.$$

Here, $\text{DTW}_{i,j}$ measures the mean similarity along the DTW-optimal path for pair $j$ at layer $i$, $\text{BG}_{i,j}$ normalizes against similarities with non-aligned text tokens, and $\text{SS}_i$ means Similarity Score of layer i quantifies the relative strength of text–speech alignment at Transformer layer $i$, with $\lambda$ serving as a global coefficient averaged over all pairs and layers.

## F    LAYER-WISE UNFREEZE LEARNING RATE SCHEDULE

To implement gradual layer-wise unfreezing, we assign each transformer layer its own learning rate with a *delayed warmup–cosine* schedule. Let $s$ denote the global training step, the model contain $N = 32$ layers to be unfrozen indexed by $i \in \{0, \ldots, N-1\}$ (with $i = N-1$ denoting the final layer), and define:

$$d_i = (N - 1 - i)\,k, \quad D_i = T - d_i - w,$$

where $k$ is the per-layer delay (in steps), $T$ the global step at which all layers have reached $\eta_{\min}$, $\eta_{\max}$ the peak learning rate, and $r = 0.1$ the final decay ratio.

For each layer $i$, we define $u = s - d_i$ and set its learning rate as

$$\eta_i(s) = \begin{cases} 0, & u < 0, \\[2mm] \eta_{\max}\,\dfrac{u}{w}, & 0 \le u < w, \\[2mm] \eta_{\min} + \dfrac{\eta_{\max} - \eta_{\min}}{2}\left(1 + \cos\left(\pi\,\dfrac{u-w}{D_i}\right)\right), & w \le u \le w + D_i, \\[2mm] \eta_{\min}, & u > w + D_i\,. \end{cases}$$

**Variable definitions**

- $s$: global training step.
- $i$: layer index ($N - 1$ = last layer).
- $N$: total number of layers (32).
- $k$: per-layer delay (5000 steps).
- $w$: warmup duration (2000 steps).
- $T$: global step when all layers reach $\eta_{\min}$.
- $\eta_{\max}$: maximum learning rate.
- $\eta_{\min} = 0.1\,\eta_{\max}$: minimum learning rate.
- $d_i = (N - 1 - i)\,k$: start delay for layer $i$.
- $D_i = T - d_i - w$: cosine decay duration for layer $i$.

This schedule ensures that higher (later) layers are unfrozen earlier, while lower (earlier) layers remain frozen longer, enabling a controlled and stable adaptation of the pretrained text backbone.

## G    RELATED WORKS

**Codec**    Speech codecs are crucial for speech large language models (SLMs) and can be grouped into two categories. Neural acoustic codecs based on (R)VQ-GAN optimize reconstruction loss to preserve fine-grained acoustic details (Zeghidour et al., 2021; Défossez et al., 2022; Kumar et al., 2023), but their tokens often lack semantic coherence when used for language modeling. In contrast, semantic-oriented codecs adopt a single-layer VQ to encode linguistic content and recover timbre with generative modules such as conditional flow matching (Du et al., 2024; Zeng et al., 2024b). While trading off perfect fidelity, this design yields tokens better suited for semantic modeling. We therefore follow the latter approach and further enhance it with streaming encoder–decoder modules for real-time interaction.

**Speech-to-Speech Interaction Models** Existing models mostly depend on text guidance for speech generation. For instance, SpeechGPT (Zhang et al., 2023a) integrates large language models with discrete speech representations but requires text-based prompts to guide speech generation. Similarly, Moshi (Défossez et al., 2024) employs a full-duplex spoken dialogue framework, generating speech tokens from a neural audio codec, yet it still necessitates text instructions for generating speech responses. Qwen-Audio (Chu et al., 2024) accepts diverse audio inputs and outputs text, relying on textual prompts for speech understanding. LLaMA-Omni (Fang et al., 2025) and Freeze-Omni (Wang et al., 2025b) extend LLMs to process speech inputs and generate speech outputs directly, but they continue to depend on text prompts to guide the interaction. Mini-Omni (Xie & Wu, 2024) fine-tunes language models to generate text and speech responses simultaneously using instruction datasets, yet the quality of both text and speech responses is limited without prior speech pre-training. GLM-4-Voice (Zeng et al., 2024b) further advances toward speech-to-speech interaction by integrating speech input and output with large language models, but it still fundamentally relies on textual supervision for alignment and instruction following. These models demonstrate progress toward speech-to-speech interaction but still require text guidance for effective performance.

**Frozen and Progressive Training** Recent work on integrating speech into decoder-only LLMs has emphasized retaining text capabilities while extending to new modalities. A common strategy is to freeze most LLM parameters and train lightweight adapters. For instance, Wang et al. (2025b) proposed *Freeze-Omni*, which augments a frozen LLM backbone with speech encoder and decoder modules, while Das et al. (2025) introduced *SpeechVerse*, combining frozen speech and text backbones with adapters to enable zero-shot speech processing from text instructions. Beyond such frozen-backbone designs, other works adopt progressive or staged adaptation. Xie & Wu (2024) presented *Mini-Omni*, which proceeds in phases: first learning speech adapters with the LLM frozen, then performing LM-only fine-tuning to align modalities, and finally unfreezing all but the audio encoder for joint multimodal training. Together, these studies show that freezing LLM backbone helps preserve language modeling ability, while progressive unfreezing provides a pathway for more flexible and effective multimodal integration.

# H DOUBLE-BLIND HUMAN EVALUATION

We conducted an additional double-blind human evaluation specifically targeting non-verbal speech generation.

## H.1 HUMAN EVALUATION SETUP

We recruited nine independent anonymous raters, all graduate students with bachelor's degrees and CET-6 English proficiency certification, ensuring sufficient linguistic competence and evaluation reliability. None of the raters were involved in this project.

The evaluation focused on accuracy and naturalness across three controlled non-verbal behaviors:

- **Silence**: Models paused for randomly sampled durations (1–10 seconds) before responding.
- **Vocal fillers**: Models produced paralinguistic cues (e.g., light laughter or sighs) prior to verbal responses.
- **Response style**: Models replied in designated affective states (e.g., hesitant or confident).

## H.2 EVALUATION PROTOCOL

Each condition included five distinct prompts. Raters evaluated every model response using a 5-point Likert scale along three dimensions:

- **Speech Naturalness**
  - 5: completely natural (human-indistinguishable)
  - 4: natural with minor flaws

- – 3: acceptable with noticeable defects
- – 2: unnatural with clear issues
- – 1: severely unnatural / unintelligible

- **Instruction Adherence**
  - – 5: perfect compliance
  - – 4: minor deviations
  - – 3: partial compliance
  - – 2: minimal compliance
  - – 1: non-compliant

- **Response Quality**
  - – 5: high relevance / accuracy
  - – 4: trivial errors
  - – 3: moderately acceptable
  - – 2: poor but marginally reasonable
  - – 1: irrational or strongly non-human

Final MOS values were computed by averaging across all raters and dimensions.

## H.3 RESULTS

Our model demonstrates substantial improvements, particularly in fine-grained prosodic behaviors such as pauses and paralinguistic vocalizations. Table 8 summarizes the MOS results.

| Non-Verbal Behavior | Ours | MIMO | GLM-4 | Kimi | Qwen3 | Gemini | GPT-4o |
|---|---|---|---|---|---|---|---|
| Silence | **4.17** | 2.40 | 2.35 | 1.93 | 2.54 | 2.73 | 2.81 |
| Vocal fillers | **4.15** | 3.85 | 3.04 | 3.56 | 3.04 | 2.85 | 3.11 |
| Response style | **4.25** | 3.60 | 3.81 | 3.85 | 3.44 | 3.22 | 3.59 |

Table 8: MOS results for non-verbal behavior generation across models.

These results highlight clear differences in non-verbal expressiveness across models. Our system consistently excels across all three categories—especially in fine-grained prosodic control—indicating strong capacity for natural, interpretable, and controllable non-verbal behavior. We believe this human evaluation provides concrete empirical evidence that directly addresses concerns that UTMOS and WER alone do not capture prosodic quality.

