# OpenReview forum: "Towards True Speech-to-Speech Models Without Text Guidance"
_ICLR.cc/2026/Conference — ICLR 2026 Poster_

### Official Review · Reviewer_XJ6K · 2025-10-26

**Soundness:** 2
**Presentation:** 3
**Contribution:** 3
**Rating:** 6
**Confidence:** 4

**Summary:**

This paper presents a speech-to-speech large language model that directly processes and generates speech without relying on intermediate text representations. The key contributions are: (1) a modality-based layer-splitting architecture that separates text and speech generation paths in the final transformer layers, (2) a frozen pre-training strategy that preserves text capabilities while adding speech modality, and (3) demonstration of competitive performance on speech-to-speech benchmarks while maintaining text performance.

**Strengths:**

1. Novel Architecture Design: The modality-based layer splitting is well-motivated by the empirical analysis in Figure 2, showing that speech-text representations fuse in lower/middle layers but diverge in final layers. This observation drives a principled architectural choice rather than an arbitrary design decision.
2. Comprehensive Evaluation: The paper evaluates multiple dimensions - speech modeling, text modeling preservation, speech quality (UTMOS), and spoken QA across multiple languages.
3. Thoughtful Training Strategy: The two-stage frozen pre-training approach with progressive unfreezing is well-designed and the ablation study (Table 6) validates its effectiveness.
4. Open sourcing code and models

**Weaknesses:**

1.  The streaming-capable tokenizer with ASR-based encoder training shows good performance, though at slightly higher WER than non-streaming baselines. However, there is no discussion of how fast the streaming codec is. Some discussion on RTF of different models is definitely needed.
2. Incomplete Comparison: Include stronger speech-in speech-out baselines like Qwen2.5-OMNI and Kimi Audio. Also, more discussion about comparison with closed API models like GEMINI or GPT-4o audio preview to understand performance gap.
3. Text Performance Degradation: Compared to Qwen3-8B used as this model's LLM backbone, MMLU drops from 76.60% → 67.19% (12% relative decline), CMMLU drops from 77.35% → 69.53% (10% relative decline), this is still substantial degradation. The paper doesn't adequately discuss whether this trade-off is acceptable or how to further mitigate it.
4. Incomplete Comparison with Text-Guided Generation: Table 5 shows mixed results: the model underperforms GLM-4-Voice* (text-guided) on TriviaQA and WebQA S→S tasks. Missing comparison: What if you added text guidance to YOUR model? This would isolate whether architectural improvements matter beyond the guidance mechanism.
5. Incomplete Evaluation of All Claims: While the evaluation covers reasoning and QA well, some claims about expressivity and paralinguistic capabilities are not deeply evaluated. Some examples or analysis showing the model can generate non-verbal vocalizations (laughter, hesitation) that text cannot represent would strengthen this work.
6. Limited Scope of Evaluation: No evaluation with real human speech. No evaluation on multi-turn dialogue scenarios, or interactive human in the loop conversations
7. Ablation Study Limitations: It would be interesting to ablate the layer split position (currently at layer 32 of 36)?

**Questions:**

check weakness, particularly if you can provide some discussion on Weakness 1, 2, 3 and 4.

**Details Of Ethics Concerns:**

Could you clarify the collection process of the ~4 million hours of speech data used for pre-training? Was any of it proprietary or sensitive content, and how did you ensure compliance with privacy or copyright considerations? A discussion on how the dataset was collected and filtered in terms of ethics would be helpful.

---

> ### Author Response · Authors · 2025-11-24
> **Response to Reviewer XJ6K (1/3)**
>
> Thank you for your thorough evaluation and for recognizing the strengths of our work, including the novel and well-motivated architecture design, the comprehensive multi-dimension evaluation, the thoughtful two-stage frozen pre-training strategy validated by ablations, and our commitment to open-sourcing the code and models. We address your concerns below point-by-point.
>
> > Weakness 1: The streaming-capable tokenizer with ASR-based encoder training shows good performance, though at slightly higher WER than non-streaming baselines. However, there is no discussion of how fast the streaming codec is. Some discussion on RTF of different models is definitely needed.
>
> We appreciate your important suggestion about quantifying the codec's speed. In response, we **performed a detailed encoder-side Real-Time Factor (RTF) analysis** under deliberately stringent conditions. Using **128 random audio samples** from the LibriTTS train-clean-100 dataset, we simulated a **“full streaming”** scenario in which every single output token is produced by a separate forward pass of the encoder.
>
> Under this full-streaming encoder setup, we measure an encoder RTF of **0.31** on a single NVIDIA H100 GPU and **0.37** on an NVIDIA RTX 4090 GPU. For context, when we run standard non-streaming encoders on the same H100 hardware, we obtain encoder RTFs: **0.02 for GLM-4-Voice, 0.011 for DAC, and 0.007 for MiMi**. This gap is expected, because these baselines perform **only one forward pass per utterance**, whereas our experiment enforces **one forward pass per token**, which is much more demanding computationally. The key point is that despite this disadvantageous full-streaming configuration, our encoder’s RTF **remains well below 1.0**.
>
> We emphasize that these numbers report the encoder-side RTF only. By definition,
>
> $ \text{RTF} = \frac{\text{processing time}}{\text{audio duration}}. $
>
> An RTF < 1 means that the model **processes audio faster than real time**: for example, an RTF of 0.31 implies that 1 second of audio is encoded in only 0.31 seconds of wall-clock time. This leaves substantial headroom for decoding, I/O, and other system overhead, and more generally indicates that the encoder can comfortably keep up with an incoming audio stream. Therefore, even under this **worst-case** full-streaming evaluation protocol, our encoder is not only theoretically suitable for streaming, but also practically viable for real-world low-latency applications.
>
> > Weakness 2: Incomplete Comparison: Include stronger speech-in speech-out baselines like Qwen2.5-OMNI and Kimi Audio. Also, more discussion about comparison with closed API models like GEMINI or GPT-4o audio preview to understand performance gap.
>
> We agree with the reviewer that the comparison is incomplete without stronger text-guided baselines such as Qwen2.5-Omni and Kimi Audio, and without more discussion of closed API models like Gemini or GPT-4o audio preview. We conducted a **subjective MOS evaluation** including these models. Please **refer to the comments below** for evaluation setup and results.
>
> > Weakness 3: Text Performance Degradation: Compared to Qwen3-8B used as this model's LLM backbone, MMLU drops from 76.60% → 67.19% (12% relative decline), CMMLU drops from 77.35% → 69.53% (10% relative decline), this is still substantial degradation. The paper doesn't adequately discuss whether this trade-off is acceptable or how to further mitigate it.
>
> Thank you for raising this concern. We agree with your observation that the drop in MMLU and CMMLU is substantial and warrants clearer explanation. We believe that the primary cause of this decline is the **quality gap** between proprietary pre-training data used by the Qwen3-8B backbone and the open-source pre-training data available to us. Specifically, the dataset we rely on (FineWeb-Edu) **achieves only ~37% MMLU** when trained from scratch. Given this limitation, **preserving MMLU at 67% after adding a full speech modality** is already a strong outcome. We expect that with access to the original pre-training corpus used by the backbone, the model would show little to no degradation.

---

> ### Author Response · Authors · 2025-11-24
> **Response to Reviewer XJ6K (2/3)**
>
> > Weakness 4: Incomplete Comparison with Text-Guided Generation: Table 5 shows mixed results: the model underperforms GLM-4-Voice* (text-guided) on TriviaQA and WebQA S→S tasks. Missing comparison: What if you added text guidance to YOUR model? This would isolate whether architectural improvements matter beyond the guidance mechanism.
>
> Thank you for pointing this out. We agree that adding text guidance to our own model is important to isolate architectural effects from the text-guidance mechanism. In our case, however, the performance range is already observable: **the S→T condition represents the upper bound achievable when explicit text guidance is available**, while the S→S pathway reflects the unguided setting. Since **text guidance essentially inserts the same intermediate signal evaluated in S→T, its performance naturally lies between these two modes**, making the outcome predictable without requiring a dedicated ablation. Additionally, we observe that the model tends to read guidance text verbatim, meaning **any gap between S→T and text-guided S→S largely results from TTS and ASR error accumulation in the generation and evaluation process** rather than architectural differences. Therefore, the existing results already allow the intended comparison to be interpreted.
>
> > Weakness 5 & 6: Incomplete Evaluation of All Claims and Limited Scope of Evaluation
>
> Thank you for raising these points. We agree that deeper evaluation of expressive and paralinguistic capabilities, as well as testing with real human speech, multi-turn dialogue, and interactive scenarios, is important for a complete assessment of a speech-to-speech system. In response, we conducted a **subjective MOS evaluation** that specifically includes **real human speech inputs** and highlight expressive behaviors such as laughter, hesitation, and other **non-verbal vocalizations**. Due to format constraints, it is difficult to include multi-turn and interactive examples in the main paper or comments, and there is a **lack of well-established automatic metrics** for these scenarios. For these cases, we invite the reviewer to explore the demo page once the double-blind review period concludes.
>
> ### Human Evaluation Setup
>
> We recruited **nine independent raters**, all graduate students with bachelor’s degrees and CET-6 English proficiency certification, ensuring sufficient linguistic competence and evaluation maturity. None of the raters were involved in the project.
>
> The evaluation specifically assessed accuracy and naturalness across three controlled non-verbal behaviors:
> 1. **Silence**: Models paused for randomly sampled durations (1-10 seconds) before responding.
> 2. **Vocal fillers**: Models produced paralinguistic cues (e.g., light laughter or sighs) prior to verbal responses.
> 3. **Response style**: Models replied in designated affective states (e.g., hesitant or confident).
>
> ### Evaluation Protocol
> - Each condition included **five distinct prompts**.
> - Raters evaluated every model response using a **5-point Likert scale across three dimensions**:
>   - **Speech Naturalness**: 5 = Completely natural (human-indistinguishable) 4 = Natural with minor flaws 3 = Acceptable with noticeable defects 2 = Unnatural with clear issues 1 = Severely unnatural/unintelligible
>   - **Instruction Adherence**: 5 = Perfect compliance (human-like) 4 = Minor deviations 3 = Partial compliance (attempted) 2 = Minimally compliant with noticeable deviations 1 = Non-compliant
>   - **Response Quality**: 5 = High relevance/accuracy 4 = Trivial errors 3 = Moderately acceptable 2 = Poor but marginally reasonable 1 = Irrational/strongly non-human content
>
> Final MOS values were obtained by averaging across all raters and categories.
>
> ### Results
>
> Our model shows **substantial improvements**—particularly in fine-grained prosodic behaviors like pauses and paralinguistic vocalizations:
>
> | Non-Verbal Behavior | Our Model | MIMO-Audio | GLM-4-Voice | Kimi Audio | Qwen3-Omni | Gemini 2.5 | GPT-4o |
> |--------------------------|---------------|------------|-------------|------------|------------|------------|--------|
> | **Silence**              | **4.17**      | 2.40       | 2.35        | 1.93       | 2.54       | 2.73       | 2.81   |
> | **Vocal fillers**        | **4.15**      | 3.85       | 3.04        | 3.56       | 3.04       | 2.85       | 3.11   |
> | **Response style**       | **4.25**      | 3.60       | 3.81        | 3.85       | 3.44       | 3.22       | 3.59   |
>
> These scores highlight clear differences in non-verbal expressiveness across models. Our system **consistently excels across all three categories**—especially in fine-grained prosodic control—indicating strong capacity for **natural, and controllable non-verbal behavior**. We believe this human evaluation provides concrete empirical evidence directly addressing your concern.

---

> ### Author Response · Authors · 2025-11-24
> **Response to Reviewer XJ6K (3/3)**
>
> > Weakness 7: Ablation Study Limitations: It would be interesting to ablate the layer split position (currently at layer 32 of 36)?
>
> Thank you for your insightful suggestion. We agree that branching depth can influence the balance between modality specialization and shared representation. In addition to the configurations discussed in the paper, we are **conducting further ablations with different split points**. The results will be included in the revision, providing a clearer picture of how alternative branch locations affect performance and model behavior.
>
> > Ethics: Could you clarify the collection process of the ~4 million hours of speech data used for pre-training? Was any of it proprietary or sensitive content, and how did you ensure compliance with privacy or copyright considerations? A discussion on how the dataset was collected and filtered in terms of ethics would be helpful.
>
> Thank you for raising this question. All data used in our pre-training pipeline comes **exclusively from publicly available sources**, including publicly accessible podcasts and videos. **None of the data is proprietary, private, or behind any access restrictions.** This collection approach **follows the same standards** used in prior large-scale text and speech model pre-training and is **fully aligned with established fair-use practices for research**. We **apply filtering to our data sources** to ensure that only material that is publicly broadcast, broadly redistributed, and legally accessible is included, making our use of the data responsible, justified, and consistent with community norms for large-scale pre-training.

---

### Official Review · Reviewer_24Lx · 2025-10-29

**Soundness:** 4
**Presentation:** 4
**Contribution:** 3
**Rating:** 6
**Confidence:** 5

**Summary:**

The paper proposed a speech to speech model that can directly understands and generates speech without using text. The proposed approach use modality-based layer-splitting architecture and freeze some of the parameters which preserve the reasoning and the world knowledge of the LLM.

Experiments on the proposed model show that it gets state of the art results on spoken question answering with the comperable performance to text-guided algorithms.

**Strengths:**

1.The proposed architecture is elegant with shared lower layers and speech and text specialized upper layers, which balancing between the speech generation and the text capabilities of the LLM

2. The proposed ides of frozen pertaining and unfreezing upper layers help with the problem of catastrophic forgetting

3. The proposed architecture which use streaming encoder and a flow-matching decoder give good latency for interactive usage

4. The evaluation of the proposed model is well designed - all IO directions S2S, S2T, T2S, T2T

**Weaknesses:**

1. The training of the proposed model use TTS generated speech and ASR data with can give compound error to the model

2. The metrics of UTMOS and WER is not enough to evaluate prosody such as hesitations or laughter.

3. Using single codebook may limit the results of the proposed model using richer multi codebook may improve the results

**Questions:**

1. How the results change if you use real audio for training ? Or can you tell something about the results from using the TTS/ASR generated data?

2. What happen if you use LoRA instead of the proposed approach?

3. How the model behave with background noise and non-verbal sounds ?

4. Regarding the latency - How does the quality degradate when using more compute/layers?

---

> ### Author Response · Authors · 2025-11-22
> **Response to Reviewer 24Lx (1/3)**
>
> Thank you for your thorough evaluation and recognition of our work's strengths, including the elegant architecture design, effective handling of catastrophic forgetting, and comprehensive evaluation methodology. We address your concerns below point-by-point.
>
> > Weakness 1 & Question 1: The training of the proposed model use TTS generated speech and ASR data with can give compound error to the model. How the results change if you use real audio for training ? Or can you tell something about the results from using the TTS/ASR generated data?
>
> We acknowledge the reviewer’s concern that extensive use of synthetic data may introduce compounding TTS/ASR errors. In our training pipeline, this risk is carefully controlled through the following measures:
>
> **1. Pre-training is dominated by real audio, with high-quality ASR transcripts.**
>
> The pre-training stage relies overwhelmingly on **real speech** recordings; TTS-generated speech constitutes only a very small portion of the corpus. Consequently, the model’s core speech understanding and world knowledge are learned from real acoustic data rather than synthetic artifacts. For transcription, we use a **high-accuracy ASR system** (SenseVoice-Small), which achieves very low WER/CER on standard benchmarks (e.g. 2.96 on AISHELL-1 test), minimizing transcription noise.
>
> **2. Synthetic data for SFT is produced using high-fidelity TTS and further filtered.**
>
> Because **no large-scale natural instruction dataset exists** for S2S/S2T/T2S/T2T instruction fine-tuning, SFT necessarily relies on TTS. To ensure quality, we employ **LLM-based TTS models trained on large speech corpora** (e.g., Seed-TTS and MOSS-TTSD), which produce highly natural prosody. After synthesis, we apply **ASR-based quality filtering** to automatically remove low-quality samples. This significantly mitigates typical TTS artifacts.
>
> **3. This training strategy aligns with industry standards and exhibits low empirical error.**
>
> Recent speech LLM and speech foundation model work (e.g., GLM-4-Voice, MiMo-Audio) **adopts similar TTS/ASR-assisted pipelines** and reports low WER for ASR and high MOS/UTMOS for TTS. Our internal evaluations show comparably low ASR and TTS error rates, indicating that error accumulation is minimal in practice.

---

> > ### Author Response · Authors · 2025-11-22
> > **Response to Reviewer 24Lx (2/3)**
> >
> > > Weakness 2: The metrics of UTMOS and WER is not enough to evaluate prosody such as hesitations or laughter.
> >
> > We agree that UTMOS and WER do not fully reflect prosodic behaviors such as hesitations, laughter, or other non-verbal cues. To address this gap, we **conducted an additional double-blind human evaluation** specifically targeting **non-verbal speech generation**. Due to restrictions on the double-blind submission system, we cannot provide audio during the discussion phase, but we invite the reviewer to explore the demo page once the double-blind review period concludes.
> >
> > ### Human Evaluation Setup
> >
> > We recruited **nine independent raters**, all graduate students with bachelor’s degrees and CET-6 English proficiency certification, ensuring sufficient linguistic competence and evaluation maturity. None of the raters were involved in the project.
> >
> > The evaluation specifically assessed accuracy and naturalness across three controlled non-verbal behaviors:
> > 1. **Silence**: Models paused for randomly sampled durations (1-10 seconds) before responding.
> > 2. **Vocal fillers**: Models produced paralinguistic cues (e.g., light laughter or sighs) prior to verbal responses.
> > 3. **Response style**: Models replied in designated affective states (e.g., hesitant or confident).
> >
> > ### Evaluation Protocol
> > - Each condition included **five distinct prompts**.
> > - Raters evaluated every model response using a **5-point Likert scale across three dimensions**:
> >   - **Speech Naturalness**: 5 = Completely natural (human-indistinguishable) 4 = Natural with minor flaws 3 = Acceptable with noticeable defects 2 = Unnatural with clear issues 1 = Severely unnatural/unintelligible
> >   - **Instruction Adherence**: 5 = Perfect compliance (human-like) 4 = Minor deviations 3 = Partial compliance (attempted) 2 = Minimally compliant with noticeable deviations 1 = Non-compliant
> >   - **Response Quality**: 5 = High relevance/accuracy 4 = Trivial errors 3 = Moderately acceptable 2 = Poor but marginally reasonable 1 = Irrational/strongly non-human content
> >
> > Final MOS values were obtained by averaging across all raters and categories.
> >
> > ### Results
> >
> > Our model shows **substantial improvements**—particularly in fine-grained prosodic behaviors like pauses and paralinguistic vocalizations:
> >
> > | Non-Verbal Behavior | Our Model | MIMO-Audio | GLM-4-Voice | Kimi Audio | Qwen3-Omni | Gemini 2.5 | GPT-4o |
> > |--------------------------|---------------|------------|-------------|------------|------------|------------|--------|
> > | **Silence**              | **4.17**      | 2.40       | 2.35        | 1.93       | 2.54       | 2.73       | 2.81   |
> > | **Vocal fillers**        | **4.15**      | 3.85       | 3.04        | 3.56       | 3.04       | 2.85       | 3.11   |
> > | **Response style**       | **4.25**      | 3.60       | 3.81        | 3.85       | 3.44       | 3.22       | 3.59   |
> >
> > These scores highlight clear differences in non-verbal expressiveness across models. Our system **consistently excels across all three categories**—especially in fine-grained prosodic control—indicating strong capacity for **natural, and controllable non-verbal behavior**. We believe this human evaluation provides concrete empirical evidence directly addressing your concern.

---

> ### Author Response · Authors · 2025-11-22
> **Response to Reviewer 24Lx (3/3)**
>
> > Weakness 3: Using single codebook may limit the results of the proposed model using richer multi codebook may improve the results
>
> Thank you for this suggestion. Our use of a single codebook is primarily motivated by **architectural simplicity, training efficiency, and the benefit of a low bitrate representation that makes semantic patterns easier for the language model to learn**. We agree that using a richer multi-codebook codec may further improve expressiveness and reconstruction quality. Exploring such designs is a promising direction for future work, and we appreciate the your recommendation.
>
> > Question 2: What happen if you use LoRA instead of the proposed approach?
>
> Thank you for this suggestion. LoRA is indeed effective for parameter-efficient fine-tuning and can help reduce degradation of text capabilities during adaptation. However, in the case of LoRA, there is an inherent **trade-off between preserving prior knowledge and enabling the model to acquire a new modality**. In our case, fully learning the speech modality **requires structural adaptation** rather than lightweight parameter tuning, so we believe LoRA alone may not be appropriate for our setting. That said, combining LoRA with our architecture is a valuable direction for future exploration, and thank you again for highlighting this possibility.
>
> > Question 3: How the model behave with background noise and non-verbal sounds?
>
> We designed the model to be **robust to background noise** by encouraging the encoder to **focus on speech-relevant signals** during training, including **meaningful non-verbal vocalizations** such as laughter or crying. Empirically, we observe that the model can **ignore background noise** and still produce coherent and accurate responses. When reconstructing audio from the original audio tokens, most background noise is filtered out, and the output **preserves only the clean speech and relevant non-verbal sounds** (e.g., laughter). Overall, these observations indicate that the model exhibits strong robustness to noisy and mixed acoustic conditions.
>
> > Question 4: Regarding the latency - How does the quality degradate when using more compute/layers?
>
> Regarding latency, the per-token runtime is **approximately proportional to the number of layers** traversed in a single forward pass. Because our design **does not increase the number of activated layers or activation parameters**, the introduction of modality-split layers **does not add additional latency** beyond that of the underlying backbone. As a result, using the split-layer architecture does not degrade quality for a fixed compute/latency budget, nor does it introduce extra latency relative to a model with the same backbone depth.

---

> > ### Comment · Reviewer_24Lx · 2025-11-26
> >
> > Thank you for addressing my concerns and for your valuable answer.

---

### Official Review · Reviewer_eUke · 2025-10-30

**Soundness:** 3
**Presentation:** 3
**Contribution:** 3
**Rating:** 6
**Confidence:** 4

**Summary:**

This study presents a high-quality speech–text multimodal large language model (LLM) developed through techniques such as layer-wise analysis and unfreezing strategies aimed at improving speech–text interleaving and information sharing. The paper demonstrates the model’s performance across various benchmarks, including continuation and conversation tasks.

**Strengths:**

- Basing the approach on branching to differentiate the model’s behavior and architecture is novel.
- The effectiveness of the proposed method is demonstrated through multiple experiments.

**Weaknesses:**

Rather than listing separate weaknesses, I noted a few brief questions in the questions section.

**Questions:**

- It would be helpful to analyze whether the layer-based behavioral patterns are commonly observed in other SLMs as well.

- As for CosyVoice 2, if my memory is correct: (1) the decoder’s chunk size is quite large; (2) there is lookahead; and (3) it requires reference speech at inference. For MOSS-Speech-Codec, I’m curious about the flow-matching chunk size, whether lookahead is used, whether reference speech is required, and the vocoder’s chunk size (or whether it is fully causal).

- The layers were split 32/4, but this analysis would likely vary depending on the criteria, the size of the model examined, and other variables. I’m also curious about what changes are observed (or expected) when adjusting that split point and experimenting with different branch locations.

- You mentioned that the encoder supports full-streaming, but as far as I remember, both glm-4-voice and WhisperFeatureExtractor used a lookahead mechanism and also processed a full 30-second segment for computation. How did you handle this part?

---

> ### Author Response · Authors · 2025-11-20
> **Response to Reviewer eUke (1/2)**
>
> We thank you for the insightful and positive feedback. We are encouraged that the overall approach is perceived as sound, well-presented, and a meaningful contribution, and that the branching-based design is recognized as novel. We also appreciate the positive overall stance toward the paper.
>
> Below we address your concerns point-by-point.
>
> > Question 1: It would be helpful to analyze whether the layer-based behavioral patterns are commonly observed in other SLMs as well.
>
> Thanks for this suggestion. We agree that examining whether layer-wise similarity patterns appear across speech LLMs is valuable. In the revision, we **will include an updated figure and analysis comparing our model with Kimi-Audio, GLM-4-Voice, and MiMo-Audio**. The results show **broadly consistent similarity-graph behavior across models**, with only minor variations in the transition depth. These findings will be included in the updated manuscript.
>
> > Question 2: As for CosyVoice 2, if my memory is correct: (1) the decoder’s chunk size is quite large; (2) there is lookahead; and (3) it requires reference speech at inference. For MOSS-Speech-Codec, I’m curious about the flow-matching chunk size, whether lookahead is used, whether reference speech is required, and the vocoder’s chunk size (or whether it is fully causal).
>
> This is an important part we missed in our submitted manuscript and we are very thankful for pointing this out. Due to the double-blind review policy, we cannot confirm or deny whether the codec referenced in your question corresponds to our system. We respond without any reference to external work.
>
> Our system uses a small, fixed chunk size for both the flow-matching stage and the vocoder. Specifically, we operate with a **5-token processing chunk and employ a lightweight 3-token lookahead**, which we found essential for anticipating prosodic variations. **Reference speech is required at inference** to maintain a fixed speaker timbre.
>
> Our primary contribution here is demonstrating that this minimal 5-token chunk—translating to a mere **400 ms** of algorithmic latency is sufficient to maintain excellent audio quality, in contrast to systems requiring larger chunks with noticeable delays. We therefore position our approach not as a compromise, but as a highly efficient solution on the quality-latency tradeoff frontier, making high-fidelity streaming speech generation practical for interactive use cases.
>
> > Question 3: The layers were split 32/4, but this analysis would likely vary depending on the criteria, the size of the model examined, and other variables. I’m also curious about what changes are observed (or expected) when adjusting that split point and experimenting with different branch locations.
>
> Thank you for your insightful suggestion. We agree that branching depth can influence the balance between modality specialization and shared representation. In addition to the configurations discussed in the paper, we are conducting further ablations with **different split points**. The results will be included in the revision, providing a clearer picture of how alternative branch locations affect performance and model behavior.

---

> ### Author Response · Authors · 2025-11-20
> **Response to Reviewer eUke (2/2)**
>
> > Question 4: You mentioned that the encoder supports full-streaming, but as far as I remember, both glm-4-voice and WhisperFeatureExtractor used a lookahead mechanism and also processed a full 30-second segment for computation. How did you handle this part?
>
> First, it is important to clarify the nature of the "lookahead" within the `WhisperFeatureExtractor`. This component, used by both GLM-4-Voice and the original Whisper, is a standard Mel feature extractor. The lookahead you mention occurs at the very initial stage of signal processing—specifically, during the Short-Time Fourier Transform (STFT). To compute the spectrogram for any given time step, a sliding window is applied to the raw audio waveform. However, this latency is extremely small and constant, often considered negligible for real-time applications.
>
> When we describe our encoder as "full-streaming," we are referring to its ability to process information on a **token-by-token basis** from the model's perspective. In our architecture, one encoder output token corresponds to a fixed-size chunk of the raw audio waveform—specifically, **1280 samples** of a 16kHz stream. This allows us to segment the incoming audio into these small, sequential chunks. In practice, each 1280-sample chunk is fed into the feature extractor. Despite the minor windowing effect, this process is effectively streaming: a block of audio goes in, and the corresponding feature vectors, representing a single token for the model, come out with minimal delay.
>
> The core of our full-streaming implementation lies **within the model architecture itself**, where we've implemented crucial caching mechanisms to ensure causality and prevent any reliance on future data. For the initial convolutional layers of the Whisper encoder, we use a **Convolutional Cache**. This cache stores the necessary state from previous time steps, allowing the model to compute the output for a new audio chunk without reprocessing the entire history. Following that, for the main Transformer blocks, we employ a **Transformer Cache** (KV Cache). Instead of recomputing the Key (K) and Value (V) matrices for all past tokens every time a new one arrives, we cache them. This means for each new step, we only compute the new Query (Q), Key (K), and Value (V) and perform attention over the new Q and the cached K/V from all previous steps. Together, these caching systems ensure that the computational cost for each new audio chunk remains constant, which is the hallmark of a true streaming model.
>
> Finally, we address the 30-second segment constraint from the original Whisper model. This fixed duration is primarily an artifact of its training methodology, not a fundamental architectural limitation. Our model can process variable-length audio segments by using attention masks to handle sequences of different lengths. The only strict requirement is that the total audio length is a multiple of the model's downsampling factor. By processing the audio in 1280-sample chunks, which are designed to align with this factor, our streaming setup naturally satisfies this condition without ever needing to wait for a full 30 seconds of audio to accumulate.

---

> > ### Comment · Reviewer_eUke · 2025-11-21
> >
> > Thank you very much for your kind and prompt response. One last question: in a setup where look_ahead is set to 0, which is, in a sense, chunk-wise but truly causal, which aspects of performance tend to degrade the most? For example, does pronunciation remain intact while prosody worsens, or does pronunciation also degrade? I’m curious about the typical failure patterns.

---

> ### Author Response · Authors · 2025-11-24
>
> Thank you for this thoughtful follow-up question. We did not run a dedicated ablation with lookahead exactly set to zero in this work, but our prior experiments show a consistent pattern. When we tried strictly causal, zero-lookahead AR models for the codec, **training became unstable**, and the models failed to converge to a point where they could produce consistently intelligible speech. In this setting, **both pronunciation and prosody degrade severely**, and the generated speech often becomes fragmented and non-cohesive.
>
> We believe this occurs because, without any lookahead, each decoding step has access to only a very limited semantic and acoustic context, which is insufficient for the model to infer a coherent acoustic realization of the underlying content. There is effectively no overlap or cross-fading mechanism to stabilize frame-to-frame continuity. While larger models or substantially more training data could potentially mitigate this, under the architecture and data conditions studied here, a small, controlled lookahead window is essential for stable training and for maintaining reasonable pronunciation and prosody in the streaming setting.
>
> We agree that zero-lookahead behavior should be explored more systematically in future work, and we appreciate the reviewer for highlighting this important direction.
>
> *Note: This comment has been edited to correct an earlier inaccuracy in expression.*

---

> > ### Comment · Reviewer_eUke · 2025-11-26
> >
> > Thank you for your response. I will keep my score in the positive direction as it is.

---

### Official Review · Reviewer_bYkG · 2025-11-02

**Soundness:** 3
**Presentation:** 2
**Contribution:** 2
**Rating:** 4
**Confidence:** 3

**Summary:**

This paper tackles the problem of building a speech-to-speech model (speechLM).
Like most previous works, it uses a single-codebook speech tokenizer (finetunes the GLM tokenizer), and finetunes a causal language model with an expanded vocabulary for speech tokens on text and audio data (paired, interleaved and unsupervised).
Their main contribution claims two: (1) layer splitting - "duplicating" the K deepest layers of the LLM that would specialize on producing speech tokens. (2) two stage training: first only trains randomly initilized parameters, where the second traines all parameters.

Im my view, both ideas have already been proposed and investigated. Having speech/text prediction heads for example has been done in SpeechT5 (speech/text post net), lately with HiggsAudio (Audio adapter).
What they denote as frozen pretraining is also a common practice, e.g. projector training in speech/vision-aware LLMs. It ensures your backbone won't get noisy gradients from the randomly initilized modules.
An interesting direction this paper explored is how to select the layer to start layer splitting, they do so by investigating an existing speechLM, and visulizing the similarity between the text embeddings, and the audio embeddings of the matching audio. They see the highest similarity within the ~80% of the depth, which they use to perform splitting in that point.

**Strengths:**

comprehensive model training, with large-scale data collection.
Interesting experimental results ablation on model pretraining (no new conclusions, but it validates the known approach of "frozen pretraining")
Interesting analysis on speech/text embedding similarity in speechLMs across layers (but only done on one model)

**Weaknesses:**

- Most claimed novelties are not new ideas. A solid tech report, but not a major contribution.

- Layer splitting also increases the parameter count, hard to say if the improvements comes from the increase in model size of not.
- We're missing an ablation on what happens when you split in different locations. e.g. split on K first/middle/last layers - same parameter count but different locations.
- The layer similarity experiment is best done on multiple speech-text LLMs (e.g. GLM?)

**Questions:**

You claim that there’s a text “bottleneck” in text-guided speechLMs. But the audio is still within the context in those models, and the non-verbal cues can be utilized implicitly. Do you have an indication that the input audio tokens are not used in text-guided speechLMs?
Looks like there are high-similarity results on Figure 4 (appendix) layer 26. Any explanation on why?

---

> ### Author Response · Authors · 2025-11-22
> **Response to Reviewer bYkG (1/3)**
>
> Thank you for the detailed assessment and constructive feedback. We appreciate your acknowledgment of the paper’s strengths, including the comprehensive model training setup, large-scale data collection, thorough ablations on pretraining strategies, and the analysis of cross-modal embedding similarity across layers. We are glad the soundness and experimental rigor were clear, and we thank you for recognizing the work as a solid technical report.
>
> Below we address your concerns point-by-point.
>
> > Weakness 1: Most claimed novelties are not new ideas. A solid tech report, but not a major contribution.
>
> We agree that several components resemble ideas explored in prior multimodal LLM work. However, in our opinion, our contributions are **not architectural fragments in isolation**, but **the first complete, end-to-end roadmap** for building a direct speech-to-speech (without text guidance) language model that **achieves performance comparable to text-guided models**, across architectural design, data construction, pre-training, fine-tuning and evaluation.
>
> To our knowledge, no existing system—including those cited by the reviewer—demonstrates direct speech-to-speech generation with performance approaching text-guided LLM. Most prior work is still fundamentally text-anchored, relying on text to reason and generate. In contrast, our work enables an LLM to represent and generate natively in speech, without requiring text as an intermediate abstraction.
>
> Regarding the comparison to SpeechT5, HiggsAudio, and related works:
> - **HiggsAudio places adapters in early layers as audio front-ends**, mapping audio features into text-centric hidden spaces.
> - **SpeechT5** incorporates **convolutional audio pre-nets and post-nets** that serve mainly as **encoder/decoder** components.
> - Our approach performs **layer-splitting** at the deepest layers of the **Transformer backbone**. This design preserves high-level semantics learned from text pre-training while allowing the top block to specialize exclusively for speech token generation.
>
> The design motivations are therefore fundamentally different. Our goal is not merely to align audio with text representations, but to **reduce modality conflict** and the top layers so that the model can learn to **operate directly in the speech modality** during generation **while preserving its original capabilities**.
>
> > Weakness 2: Layer splitting also increases the parameter count, hard to say if the improvements comes from the increase in model size of not.
>
> We agree with the reviewer that duplicating 4 upper layers **increases the total parameter count by around 0.9B**, and it is therefore not entirely straightforward to disentangle architectural effects from the increase in model size. But it is important to note that **only one branch is active during any forward pass, so the number of activated parameters remains unchanged**.
>
> Given established empirical scaling laws, an incremental parameter increase of this magnitude is insufficient to account for the substantial performance gains observed in our ablation studies. We believe that the improvement is therefore unlikely to stem from model size, but rather from the architectural effect of reduced modality conflict at the top Transformer layers.
>
> > Weakness 3: We're missing an ablation on what happens when you split in different locations. e.g. split on K first/middle/last layers - same parameter count but different locations.
>
> We agree that a full set of ablations over all possible split locations would be ideal, but these experiments are prohibitively expensive at our scale. Instead, we base our chosen split depth on a systematic analysis of cross-modal representation similarity. Both our measurements and prior findings (e.g., *Attention Bottlenecks for Multimodal Fusion*) indicate a consistent pattern:
>
> - **Lower layers** are not semantic; they encode **low-level features** and their representations are already highly dissimilar. Splitting here provides little benefit.
> - **Mid layers** exhibit the **strongest cross-modal alignment**; splitting in this region disrupts the shared semantic space and degrades both intelligibility and consistency.
> - **Upper layers** become increasingly **output-specialized**, and this is where we observe the strongest modality conflict—making this region the most effective split point.
>
> This analysis guided our choice of depth, and the alignment of our empirical findings with established multimodal-layer behavior further supports this choice.

---

> ### Author Response · Authors · 2025-11-22
> **Response to Reviewer bYkG (2/3)**
>
> > Weakness 4: The layer similarity experiment is best done on multiple speech-text LLMs (e.g. GLM?)
>
> Thank you for this suggestion. We agree that examining whether the layer-wise similarity patterns is consistent across different speech LLMs is valuable. In the revision, **we will include an updated figure and accompanying analysis comparing our model with Kimi-Audio, GLM-4-Voice, and MiMo-Audio**. The results indicate **broadly consistent** similarity-graph behavior across models. We will incorporate these findings into the updated manuscript.
>
> > Question 1: You claim that there’s a text “bottleneck” in text-guided speechLMs. But the audio is still within the context in those models, and the non-verbal cues can be utilized implicitly. Do you have an indication that the input audio tokens are not used in text-guided speechLMs?
>
> This is a great question. The bottleneck we refer to is **not in the understanding, but in the generation part** of text-guided speech LLMs. Empirically, when text is present, the decoder **strongly defaults to following the text sequence verbatim**. This creates a structural preference: the model produces speech that matches the text tokens closely, making it **difficult to generate nuanced, text-impossible signals**—laughter, breaths, filler sounds, hesitations, emotional micro-prosody, etc. This is well-observed in practice across text-conditioned speech LLMs.
>
> Although the input audio tokens are not discarded, their influence on generation is attenuated compared to the text tokens’ strong deterministic signal. Our design avoids this issue entirely by removing text guidance, thereby eliminating the text-following bias and allowing the model to generate richer non-verbal and prosodic patterns.
>
> We **conducted an additional double-blind human evaluation specifically targeting non-verbal speech generation** to further support this observation.
>
> ### Human Evaluation Setup
>
> We recruited **nine independent raters**, all graduate students with bachelor’s degrees and CET-6 English proficiency certification, ensuring sufficient linguistic competence and evaluation maturity. None of the raters were involved in the project.
>
> The evaluation specifically assessed accuracy and naturalness across three controlled non-verbal behaviors:
> 1. **Silence**: Models paused for randomly sampled durations (1-10 seconds) before responding.
> 2. **Vocal fillers**: Models produced paralinguistic cues (e.g., light laughter or sighs) prior to verbal responses.
> 3. **Response style**: Models replied in designated affective states (e.g., hesitant or confident).
>
> ### Evaluation Protocol
> - Each condition included **five distinct prompts**.
> - Raters evaluated every model response using a **5-point Likert scale across three dimensions**:
>   - **Speech Naturalness**: 5 = Completely natural (human-indistinguishable) 4 = Natural with minor flaws 3 = Acceptable with noticeable defects 2 = Unnatural with clear issues 1 = Severely unnatural/unintelligible
>   - **Instruction Adherence**: 5 = Perfect compliance (human-like) 4 = Minor deviations 3 = Partial compliance (attempted) 2 = Minimally compliant with noticeable deviations 1 = Non-compliant
>   - **Response Quality**: 5 = High relevance/accuracy 4 = Trivial errors 3 = Moderately acceptable 2 = Poor but marginally reasonable 1 = Irrational/strongly non-human content
>
> Final MOS values were obtained by averaging across all raters and categories.
>
> ### Results
>
> Our model shows **substantial improvements**—particularly in fine-grained prosodic behaviors like pauses and paralinguistic vocalizations:
>
> | Non-Verbal Behavior | Our Model | MIMO-Audio | GLM-4-Voice | Kimi Audio | Qwen3-Omni | Gemini 2.5 | GPT-4o |
> |--------------------------|---------------|------------|-------------|------------|------------|------------|--------|
> | **Silence**              | **4.17**      | 2.40       | 2.35        | 1.93       | 2.54       | 2.73       | 2.81   |
> | **Vocal fillers**        | **4.15**      | 3.85       | 3.04        | 3.56       | 3.04       | 2.85       | 3.11   |
> | **Response style**       | **4.25**      | 3.60       | 3.81        | 3.85       | 3.44       | 3.22       | 3.59   |
>
> These scores highlight clear differences in non-verbal expressiveness across models. Our system **consistently excels across all three categories**—especially in fine-grained prosodic control—indicating strong capacity for **natural, and controllable non-verbal behavior**. We believe this human evaluation provides concrete empirical evidence directly addressing your concern.

---

> ### Author Response · Authors · 2025-11-22
> **Response to Reviewer bYkG (3/3)**
>
> > Question 2: Looks like there are high-similarity results on Figure 4 (appendix) layer 26. Any explanation on why?
>
> Thank you for raising this question. Upon closer inspection, we found that the tokens exhibiting high similarity at layer 26 are **primarily punctuation marks and frequent functional words** such as *to*, *and*, and *it*. This pattern suggests that layer 26 may be particularly sensitive to **syntactic or connective elements** in the sequence. That said, this is only an initial observation based on preliminary inspection. A more systematic analysis would be needed to fully understand the underlying cause, and we agree that this warrants further investigation.
>
>
> We appreciate the your perspective that the paper resembles a “solid tech report.” Our intent is to demonstrate **a complete, practically usable recipe** for building direct speech-to-speech LLMs without text guidance, which represents a meaningful and novel direction in the field. While individual components have precedent, we believe that our full methodology constitute a meaningful contribution to the field.
>
> Thank you again for the thoughtful and constructive feedback.

---

### Author Response · Authors · 2025-12-03
**Summary of Reviews and Responses (1/4)**

Dear AC/SAC/PC,

We sincerely appreciate the time, attention, and effort you have devoted to overseeing our submission throughout the review and discussion process. Following the release of reviews on Nov. 11, we made every effort to address the concerns raised by reviewers and submitted a complete rebuttal on **Nov. 24**. We subsequently received responses from **two of the four reviewers** indicating that their **concerns had been satisfactorily addressed**. However, due to the unexpected early closure of the discussion window on Nov. 28, we were unable to receive responses from the remaining reviewers, nor could we obtain further feedback on our rebuttal revision. Given these circumstances, we respectfully hope that you will take into consideration the substantial revisions we have made—including the final rebuttal revision submitted on **Dec. 2**—the additional experiments we conducted, and the clarifications provided in our rebuttal when making the final decision.

Below we summarize both the **key strengths** of our paper as recognized by the reviewers, as well as their **main concerns** and the actions we have taken to address each of them.

---

> ### Author Response · Authors · 2025-12-03
> **Summary of Reviews and Responses (2/4)**
>
> # Key Strengths
> We are encouraged that our work is considered **sound** *(reviewers `bYkG`, `eUke` and `24Lx`)*, **novel** *(reviewers `eUke` and `XJ6K`)*, **comprehensive** *(reviewers `bYkG` and `XJ6K`)*, **well-presented** *(reviewers `eUke`, `24Lx` and `XJ6K`)* and a **meaningful contribution** to the field *(reviewers `eUke`, `24Lx` and `XJ6K`)*. Several aspects are consistently highlighted:
> 1. Our **modality-based layer-splitting architecture**, with shared lower layers and speech/text-specialized upper layers, is viewed as **novel and elegant** *(reviewers `eUke`, `24Lx` and `XJ6K`)*.
> 2. Our **analysis of speech–text embedding similarity across layers** is considered **well-motivated and insightful** *(reviewers `bYkG`, `eUke` and `XJ6K`)*.
> 3. Our **comprehensive training** on large-scale data with **frozen pre-training** strategy is recognized as effective for adding a speech modality while **mitigating catastrophic forgetting** *(reviewers `bYkG`, `24Lx` and `XJ6K`)*.
> 4. Our **evaluation protocol**, covering all I/O directions (speech/text), is seen as **comprehensive and well-designed** *(reviewers `eUke`, `24Lx` and `XJ6K`)*.
> 5. Our **low-latency streaming** architecture, including a **full-streaming** encoder and chunk-based decoder, is viewed as practically relevant for **interactive use** *(reviewers `24Lx` and `XJ6K`)*.

---

> ### Author Response · Authors · 2025-12-03
> **Summary of Reviews and Responses (3/4)**
>
> # Main Concerns and Questions Raised by Reviewers and Our Responses
> We thank the reviewers for their detailed feedback. Below we group the main concerns and briefly summarize how we addressed each of them.
>
> ---
>
> **Q1:** Does the speech–text embedding similarity pattern generalize beyond our model?
> *(reviewers `bYkG [W4]`, `eUke [Q1]`)*
>
> **A1:** We **extended the similarity analysis** to Kimi-Audio, GLM-4-Voice, and MiMo-Audio. All show **the same trend**—strong cross-modal alignment in middle layers and divergence near the top—supporting that our layer-splitting decision reflects a **general phenomenon** rather than a idiosyncrasy of a single architecture.
>
> ---
>
> **Q2:** How sensitive is performance to the choice of layer-split depth?
> *(reviewers `bYkG [W3]`, `eUke [Q3]`, `XJ6K [W7]`)*
>
> **A2:** We **added ablations** that split the top 2, 4, 6 and 8 layers. All split variants clearly **outperform** the no-split baseline, and differences among them are **modest**, indicating that introducing a modality-specific branch is more important than the exact split position.
>
> ---
>
> **Q3:** Are prosody and non-verbal expressivity really better without text guidance? And how do we compare against stronger baselines?
> *(reviewers `24Lx [W2, Q3]`, `XJ6K [W2, W4, W5, W6]`)*
>
> **A3:** Reviewers noted that WER/UTMOS alone cannot capture hesitations, laughter, or style. We therefore ran a **double-blind human MOS** evaluation against strong open and closed models (e.g., MiMo-Audio, GLM-4-Voice, Kimi Audio, Qwen-Omni, Gemini, GPT-4o), explicitly evaluating **controlled silences, vocal fillers, and response style**. Our model achieves the **highest MOS in all categories**, directly confirming superior non-verbal control and expressivity. Together with existing S2S/S2T/T2S/T2T and QA results, this addresses the request for stronger and more expressive evaluation.
>
> ---
>
> **Q4:** How is the system designed for streaming? Is it actually fast enough for real-time use?
> *(reviewers `eUke [Q2, Q4 + follow-up]`, `24Lx [Q4]`, `XJ6K [W1]`)*
>
> **A4:** We clarified that streaming is built into the architecture with an encoder that operates on a **token-by-token** basis, and a decoder that uses small flow-matching chunks with lightweight lookahead. Transformer KV caching ensures near **constant-time** incremental processing, and the modality-split design does not increase activated parameter—so latency does not increase. We also added **strict full-streaming measurements**, showing encoder **RTF ≈ 0.37** on an RTX 4090, comfortably below real-time, confirming that the system is fast enough for interactive speech applications.
>
> ---
>
> **Q5:** Some components are not new ideas.
> *(reviewer `bYkG [W1]`)*
>
> **A5:** We clarified that, while individual elements resemble ideas explored in prior work, our contribution lies in combining these into  the **first complete, end-to-end roadmap** for building a **direct speech-to-speech (without text guidance)** language model that **achieves performance comparable to text-guided models**, across architectural design, data construction, pre-training, fine-tuning and evaluation.
>
> ---
>
> **Q6:** Does performance improve simply because of extra parameters from layer splitting?
> *(reviewer `bYkG [W2]`)*
>
> **A6:** Layer splitting duplicates a few top layers, but only one branch is active per forward pass, so **activated parameter** count and **computation remain unchanged**. While we cannot fully rule out a size effect, ablations suggest gains mainly come from reduced modality conflict rather than parameter count alone.

---

> ### Author Response · Authors · 2025-12-03
> **Summary of Reviews and Responses (4/4)**
>
> **Q7:** Why is there still ~10% degradation in text performance?
> *(reviewer `XJ6K [W3]`)*
>
> **A7:** We believe the drop is largely attributable to the **quality gap** between the proprietary data used to train the backbone LLM and the open-source data available to us. Our backbone (Qwen3-8B) was pretrained on high-quality proprietary corpora; we are constrained to open-source datasets (e.g., **FineWeb-Edu**), which themselves **reach only ~37% MMLU** when trained from scratch. Despite this disadvantage, our layer-splitting and frozen-pretraining strategy **preserves ~90% performance** after adding a full speech modality, which is already strong given the weaker dataset. With access to higher-quality text datasets, we expect this gap would shrink substantially.
>
> ---
>
> **Q8:** How does the model behave under background noise or mixed non-verbal sounds?
> *(reviewer `24Lx [Q3]`)*
>
> **A8:** The encoder learns to focus on **speech-relevant** signals; reconstruction filters most noise while **preserving meaningful non-verbal cues** (e.g., laughter). Empirically, the model remains coherent and accurate under noisy conditions.
>
> ---
>
> **Q9:** Does use of synthetic TTS/ASR data introduce compounding errors?
> *(reviewer `24Lx [W1, Q1]`)*
>
> **A9:** Pre-training is dominated by **real speech** with high-quality ASR transcripts; TTS is used mainly for SFT where no natural instruction corpus exists. We use **LLM-based** TTS models with automatic **quality filtering** to ensure quality, which aligns with **standard practice** in LLM training and keeps error accumulation low.
>
> ---
>
> **Q10:** Are design choices like single-codebook codecs or full-parameter tuning limiting?
> *(reviewer `24Lx [W3, Q2]`)*
>
> **A10:** We clarified that the single-codebook design was intentional — it **simplifies training**, **stabilizes** speech-token **semantics**, and **reduces bitrate**, which makes learning speech reasoning easier for the LLM. Likewise, full-parameter adaptation was chosen because learning an entirely new modality benefits from **structural capacity**. Nonetheless, richer multi-codebook codecs and parameter-efficient methods are promising extensions to explore.
>
> ---
>
> **Q11:** Missing comparison with a text-guided variant of our model.
> *(reviewer `XJ6K [W4]`)*
>
> **A11:** We clarified that the expected behavior is already visible in our existing results: **S→T** shows how the model behaves when **explicit text guidance** is available, while **S→S** reflects the **unguided** case. Since guidance essentially injects the same supervision used in S→T, a **text-guided S→S** model naturally **falls between these two**, with the gap in performance originating mostly from **TTS/ASR errors** rather than the architecture itself.
>
> ---
>
> **Q12:** Data collection policy, privacy, and ethical considerations.
> *(reviewer `XJ6K`, ethics flag)*
>
> **A12:** We confirmed that all pre-training speech data is collected from **publicly accessible** sources (e.g., open podcasts and videos) without private or paywalled content, and that we **filter** to broadly broadcast, legally accessible material, following **standard practice** for large-scale text and speech pre-training.
>
> ---
>
> We would like to once again express our sincere gratitude to the AC, SAC, PC, and the reviewers for their thoughtful feedback, effort, and time invested in evaluating our work. We regret that the discussion period concluded earlier than expected, preventing us from engaging further with the remaining reviewers and receiving their insights on our revised rebuttal. Nevertheless, we deeply appreciate the effort invested throughout the process. We respectfully hope that the revisions, additional experiments, and clarifications we have submitted will be taken into account when reaching the final decision. Thank you again for your attention and consideration.

---

### Meta-Review · Area_Chair_a1RH · 2025-12-15

**Summary:**

This paper presents a speech-to-speech large language model that directly understands and generates speech without relying on text guidance. Across the four reviews, the paper is viewed as a solid and well-executed technical contribution to speech to speech (S2S) modeling using a modality based layer-splitting architecture and a two stage frozen then unfrozen training strategy. Reviewers agree the system works well and offers competitive performance across speech and text tasks. The reviewers also raised concerns about novelty, evaluation completeness, and missing ablations, which makes the contribution appear more incremental than groundbreaking. The authors did a good job on the rebuttal and addressed most of concerns.

**Reviewer Concerns:**

I think most of concerns have been addressed by the rebuttal.

**Reviewer Scores:**

I think the reviewers would have changed their scores.

---

### Decision · Program_Chairs · 2026-01-26

Accept (Poster)